

# Rossby wave resonance for idealized jets on a beta-plane

Volkmar Wirth[1] and Nili Harnik[2]

[1]Institute for Atmospheric Physics, Johannes Gutenberg University Mainz, Becherweg 21, 55128 Mainz, Germany.
[2]Porter School of the Environment and Earth Sciences, Tel Aviv University, Israel

**Correspondence:** Volkmar Wirth (vwirth@uni-mainz.de)

**Abstract.** This paper analyzes Rossby wave resonance along a circumglobal midlatitude jet in the framework of the linearized inviscid barotropic vorticity equation on a zonally periodic beta-plane. Zonally symmetric Gaussian-shaped westery jets of varying amplitude and width are specified as basic states. The system is forced by pseudo-orography which varies sinusoidally in the zonal direction and which has a small meridional extent. Stationary solutions are obtained through straightforward numerical methods. The strength of resonant amplification is diagnosed by systematically varying the zonal wavenumber $s$, plotting the resulting wave amplitude as a function of $s$, and quantifying the sharpness of its peak (if existent). The numerical solutions for jet-like basic states are interpreted by reference to analytical solutions obtained for more idealized model configurations.

The analysis indicates that a jet with realistic amplitude and width may be subject to a weak form of resonance. Given that the zonal scale of a jet is much larger than its meridional scale, one may expect resonance at no more than one zonal wavenumber $s_{\text{res}}$. This resonant peak is associated with the first meridional mode, which is established through partial reflection of wave activity at the periphery of the jet flanks. The fact that a jet acts like a leaky waveguide implies that the wave amplitude remains finite even right at the resonant wavenumber. The behavior is very similar as in the classic Charney-Eliassen model, where the channel width must be chosen appropriately and where damping simulates the leakiness of the jet.

## 1 Introduction

It has long been known that Rossby waves can be subject to resonant amplification under specific conditions. To the best of our knowledge, the first to mention this phenomenon was Haurwitz (1940), who investigated normal modes of the barotropic vorticity equation on the sphere; he noted that some of these normal modes are close to stationary, and in this case stationary forcing with suitable spatial structure would lead to resonance. Somewhat later, Charney and Eliassen (1949) considered a similar problem, but on a beta-plane channel. In their model, forcing due to Northern Hemisphere orography gave rise to



stationary Rossby wave perturbations that resembled the observed ones. An important feature of their solution was the fact that a limited band of zonal wavenumbers experienced enhanced amplification due to the mechanism of resonance.

In both above quoted studies, a key ingredient for the occurrence of resonance is the fact that the model domain is zonally periodic; this allows wave activity to travel around the Earth several times in the zonal direction such that the waves can interfere with themselves. In the work of Haurwitz (1940), this was possible thanks to the spherical domain with global extent, while in the work of Charney and Eliassen (1949) this was possible thanks to the periodic channel with impermeable walls at the meridional boundaries. To the extent that one focuses on the midlatitudes, the configuration of Charney and Eliassen (1949) is arguably the more relevant one: the channel walls in that model can be considered as an idealized representation of strong zonal "waveguidability", that may occur along a circumglobal midlatitude jet (Manola et al., 2013).

Previous work suggests that Rossby wave resonance is of minor importance under current climatological conditions. The reason is that waves are usually subject to damping, and this seems to prevent any moderate or even strong form of resonance (Held, 1983). Moreover, it was recently shown that even in the complete absence of wave damping, a midlatitude jet is a rather leaky waveguide and the leakiness of a waveguide has a similar effect as wave damping (Wirth, 2020; Harnik and Wirth, 2025). In addition, jets are usually not truly circumglobal, and it appears likely that a non-circumglobal jet is less prone to resonance than a truly circumglobal jet.

Nevertheless, it seems important to have a thorough understanding of Rossby wave resonance, because the nature of resonance implies that small changes in relevant characteristics of the system — such as the basic state wind speed or the spatial structure of the forcing — may lead to large changes in wave amplitude. For instance, a shift towards resonant conditions during specific (possibly rare) episodes could lead to large wave amplitudes, which in turn may be associated with extreme weather. To the extent that the forcing stems from stationary sources (such as orography), the resonant waves are stationary, too, after reaching saturation, and this increases the potential for extreme weather. For the same reason, the mechanism of Rossby wave resonance may be important in connection with small trends due to anthropogenic climate change. In fact, it was suggested that Rossby wave resonance may be the mechanism underlying recent Northern Hemisphere weather extremes (Petoukhov et al., 2013; Kornhuber et al., 2017) and that the conditions for Rossby wave resonance may systematically change due to anthropogenic climate change (Mann et al., 2017)

It is the goal of the current paper to enhance our understanding for Rossby wave resonance in an idealized framework and, at the same time, suggest a diagnostic which is compatible with the underlying basic physics. We are going to work in the framework of the linearized barotropic vorticity equation on a beta-plane. As basic states we consider westerly Gaussian jets superimposed on a constant westerly background wind; these jets are subject to various realizations of the forcing, and the corresponding stationary solutions are obtained through straightforward numerical methods. In addition, we compare our numerical solutions with analytical solutions for more idealized model configurations, because this allows us to better understand the numerical results. In all cases we restrict attention to zonally symmetric basic states, which is in line with the idea that





a strong circumglobal jet can be a good waveguide, which in turn is a prerequisite for Rossby wave resonance. In addition, we only consider inviscid wave dynamics, because this produces resonance in its cleanest form. Obviously, one expects some
wave damping in any practical application, and this would reduce the tendency towards resonant behavior.

Our strategy to diagnose Rossby wave resonance makes use of a fundamental property of any oscillating system that may be subject to resonance: to the extent that the forcing is close to a normal mode of the free system, the forced system will show a particularly strong response. The way we realize this idea in our model framework is by using forcing with sinusoidal variation in the zonal direction, systematically varying the zonal wavenumber, and analysing the amplitude of the response.

The paper is organized as follows. First, in chapter 2 we present the model equations, sketch our numerical solution procedure, and describe in more detail our strategy to detect resonance. Section 3 then discusses analytical solutions in idealized model configurations, which will be subsequently used for the purpose of interpretation. Our key results are contained in section 4, where we present and discuss numerical solutions for jet-like basic states. Finally, we summarize our results and draw conclusions in section 5.

## 2 Barotropic model framework

Following a substantial body of previous work, we consider the linearized barotropic vorticity equation on a zonally periodic beta-plane. The relevant parameters as well as the basic states are chosen such that one obtains idealized representations of midlatitude jets on planet Earth. Simplicity of the model is considered a virtue rather than a weakness, as it allows us to "understand" (to a considerable extent) the resulting behavior; in particular we will interpret our numerical solution in terms
of specific analytical solutions.

### 2.1 Model setup

Our model domain is a rectangle of length $L_x$ and width $L_y$, extending from $x = 0$ to $x = L_x$ in the zonal direction and from $y = -L_y/2$ to $y = L_y/2$ in the meridional direction. In the entire paper, the length of the domain is set to

$$L_x = 2\pi a \cos \phi_0 \,, \tag{1}$$

where $a = 6371.2$ km denotes the radius of the Earth and $\phi_0 = 45°$N is a reference latitude. For terminological convenience we will refer to the zonal direction as "longitude" and the meridional direction as "latitude", although we stick to Cartesian geometry throughout the paper. The beta-plane approximation implies that the Coriolis parameter is given by

$$f(y) = f_0 + \beta y \,, \tag{2}$$

with $f_0 = 2\Omega \sin \phi_0$ and $\beta = 2\Omega a^{-1} \cos \phi_0$.





## 2.2   Model equations

We assume a basic state that is zonally symmetric and purely zonal, but its zonal wind $u_0(y)$ may depend on latitude. Linearizing the inviscid barotropic vorticity equation about this basic state and assuming some external stationary forcing $F'$, one obtains

$$\left( \frac{\partial}{\partial t} + u_0 \frac{\partial}{\partial x} \right) q' + q_{0y} v' = F' \, , \tag{3}$$

where $q'$ is perturbation absolute vorticity, $v'$ is the perturbation meridional wind, and

$$q_{0y} = \beta - \frac{d^2 u_0}{dy^2} \tag{4}$$

is the meridional gradient of the basic state absolute vorticity.

The forcing $F'$ is modelled through a dimensionless pseudo-orography $h'(x,y)$ as

$$F'(x,y) = -f_0 u_0 \frac{\partial h'}{\partial x} \, . \tag{5}$$

In the entire paper we only consider orography which is sinusoidal in longitude, i.e.,

$$h'(x,y) = \Re \, \hat{h}(y) \, e^{ikx} \, , \tag{6}$$

where $\Re \ldots$ denotes the real part and $\hat{h}(y)$ characterizes the meridional profile of the orography.

In most parts of this paper the meridional profile $\hat{h}(y)$ is assumed to be "meridionally thin". For the analytical treatment it is represented by a delta function like

$$\hat{h}(y) = D \, \delta(y) \, , \tag{7}$$

with $D = 500$ km, and this will be referred to as delta-forcing in the following. For our numerical solutions, (7) is replaced by

$$\hat{h}(y) = \frac{D}{\widetilde{D}} \cos^2 \left( \frac{\pi}{2\widetilde{D}} y \right) \quad \text{for } |y| \leq \widetilde{D} \, , \tag{8}$$

and zero otherwise. Unless stated otherwise we use $\tilde{D} = 500$ km, and in all our model configurations we satisfy $\widetilde{D} \ll L_y$. This guarantees that $\hat{h}(y)$ from (8) is "merdionally thin" and can be taken as approximation to the delta function. At the same time, $\tilde{D}$ is always chosen wide enough such that the non-zero part of $\hat{h}(y)$ in (8) can be represented by a fair number of grid points and, hence, properly resolved in our numerical treatment. Note that for both (7) and (8) one obtains

$$\int_{-L_y/2}^{L_y/2} \hat{h}(y) \, dy = D \, , \tag{9}$$





which means that the amplitude of the orography is equivalent in an integrated sense.

Writing $q'$ and $v'$ in terms of the perturbation streamfunction $\psi'$,

$$q' = \nabla^2 \psi' \,, \quad v' = \frac{\partial \psi'}{\partial x} \,, \tag{10}$$

and restricting attention to stationary solutions, one obtains

$$u_0 \frac{\partial}{\partial x} \nabla^2 \psi' + q_{0y} \frac{\partial \psi'}{\partial x} = -f_0 u_0 \frac{\partial h'}{\partial x} \,. \tag{11}$$

We look for solutions of the following form

$$\psi'(x,y) = \Re \, \hat{\psi}(y) \, e^{ikx} \,, \tag{12}$$

divide by $u_0$, and obtain

$$\frac{d^2 \hat{\psi}}{dy^2} + \left[ \frac{q_{0y}}{u_0} - k^2 \right] \hat{\psi} = -f_0 \hat{h} \,. \tag{13}$$

In the remainder of this paper we only consider basic states satisfying $u_0 > 0$ throughout the interior of the domain such that (13) is free of singularities.

For later reference we define the square of the stationary wavenumber

$$K_s^2 = \frac{q_{0y}}{u_0} \tag{14}$$

and the dimensionless stationary wavenumber

$$\hat{K}_s = \frac{L_x}{2\pi} \sqrt{K_s^2} \,. \tag{15}$$

For a constant basic state wind, both $K_s^2$ and $\hat{K}_s$ are constant, but for more general profiles of $u_0(y)$ they are functions of latitude. A typical mid-latitude jet satisfies $K_s \lesssim 10$ within the jet region (Wirth, 2020).

Introducing the dimensionless zonal wavenumber

$$s = \frac{L_x}{2\pi} k \,, \tag{16}$$

equation (13) can be rewritten as

$$\frac{d^2 \hat{\psi}}{dy^2} + \left( \frac{2\pi}{L_x} \right)^2 \left[ \hat{K}_s^2 - s^2 \right] \hat{\psi} = -f_0 \hat{h} \,. \tag{17}$$

Considering the value $L_x$ as given and fixed, the above equation indicates that the local character of the solution $\hat{\psi}$ outside the forcing region only depends on the function $\hat{K}_s(y)$ and the value of $s$. In particular, the solution has an oscillatory character





for latitudes where $\hat{K}_s^2 > s^2$, while is has an exponential character for latitudes where $\hat{K}_s^2 < s^2$. It follows that the solution $\hat{\psi}$ does not necessarily satisfy $\hat{\psi} = 0$ where $\hat{K}_s^2 = s^2$.

The meridional component of the linear wave activity flux is given by

$$F^{(y)} = -\overline{u'v'} \,, \tag{18}$$

where the overbar denotes the zonal average. For solutions with a fixed zonal wavenumber, this can be reformulated in terms of the perturbation streamfunction $\psi' = \Re \, \hat{\psi}(y) \exp(ikx)$ as

$$F^{(y)} = \frac{1}{2} \Re \left( ik \hat{\psi} \frac{d\hat{\psi}^*}{dy} \right) \,, \tag{19}$$

where the asterisk denotes the complex conjugate. Assuming furthermore that the zonal wavenumber $k$ is real and $\hat{\psi}(y) \propto \exp(ily)$, the last expression turns into

$$F^{(y)} = \frac{1}{2} k (\Re \, l) |\hat{\psi}|^2 \,; \tag{20}$$

in this case the meridional flux of wave activity vanishes if $l = 0$ or if $l$ is purely imaginary.

## 2.3 Boundary conditions and implications for quantization

Periodicity of the domain sets a constraint on $k$, namely that the zonal wavenumber must be quantized according to $kL_x \overset{!}{=} 2\pi s$ or

$$k \overset{!}{=} \frac{2\pi}{L_x} s \tag{21}$$

with $s = 1, 2, 3, \ldots$. The integer $s$ represents the number of entire wavelengths that fit into the domain in the zonal direction.

At both the southern and the northern boundary of the domain we posit that a certain fraction $R$ of wave amplitude is reflected, while the remaining part is transmitted. Following Harnik and Wirth (2025, their equation (18)), this can be achieved by specifying

$$\left( \frac{1-R}{1+R} \right) \frac{d\hat{\psi}}{dy} = \pm i \sqrt{\frac{q_{0y}}{u_0} - k^2} \, \hat{\psi} \quad \text{at} \quad y = \pm L_y/2 \,. \tag{22}$$

Note that this boundary condition is singular in the limit $R \to 1$, and one obtains the familiar Dirichlet condition $\hat{\psi} = 0$ for fully reflecting conditions $R = 1$ (except when the square root on the right hand side happens to be zero). In the remainder of this paper, the special model configuration with $R = 1$ will be referred to as a "reflecting periodic channel". The condition $\hat{\psi} = 0$ in this case represents another quantization constraint, namely that an integer number $n$ of half wavelengths must fit into the meridional extent of the channel, i.e.,

$$l \overset{!}{=} n \frac{\pi}{L_y} \tag{23}$$





with $n = 1, 2, 3, \ldots$.

## 2.4 Numerical solution procedure

Equations (17) and (22) represent a 1D boundary value problem, which can be solved numerically in a straightforward manner.
The differential operator on the left hand side of (17) is discretized using standard finite differences, reducing the solution for the interior grid points to the inversion of a square matrix. The boundary conditions are implemented by either (in case of $R = 1$) setting the boundary grid points of $\hat{\psi}$ to zero, or (in case of $R < 1$) by modifying the equations for the interior grid points such as to account for the discretized version of (22). The resulting square matrix is inverted using a linear algebra routine from `scipy`.

## 2.5 Diagonostic strategy

In case of the forced harmonic oscillator from theoretical physics, there is a straightforward recipe to diagnose resonant behavior: try many different values for the forcing frequency (using identical forcing amplitude) and determine whether and to what extent the stationary reponse shows a pronounced peak in amplitude in the neighborhood of a specific forcing frequency.

Our strategy to diagnose Rossby wave resonance closely follows this idea: we compute the stationary solution $\psi'$ for an
entire range of zonal wavenumbers $s$ (with the same forcing amplitude, i.e., the same value of $D$ for each value of $s$) and evaluate how different aspects of the solution change as a function of $s$. One particular "aspect" of the solution is, obviously, its amplitude: to the extent that the amplitude shows a pronounced peak at one or several specific values of $s$, we associate the basic state with resonant behavior at these values of $s$. In fact, for the current purpose we can consider $s$ to be a positive real number (rather than an positive integer), because the solution for the meridional structure problem is effectively ignorant of the
boundary conditions in the zonal direction. In addition, we analyze the solution's phase as a function of $s$, because the phase behavior serves as another hallmark of resonance (Harnik and Wirth, 2025).

In the following two section we are going to consider various model configurations that differ in the basic state zonal wind $u_0(y)$ and in the choice of the boundary conditions. For illustration the reader is refered to Fig. 1. The configurations depicted in Fig. 1a, b, and c allow analytical solutions (section 3), while the configuration in Fig. 1d requires one to resort to the numerical
solution procedure (section 4).





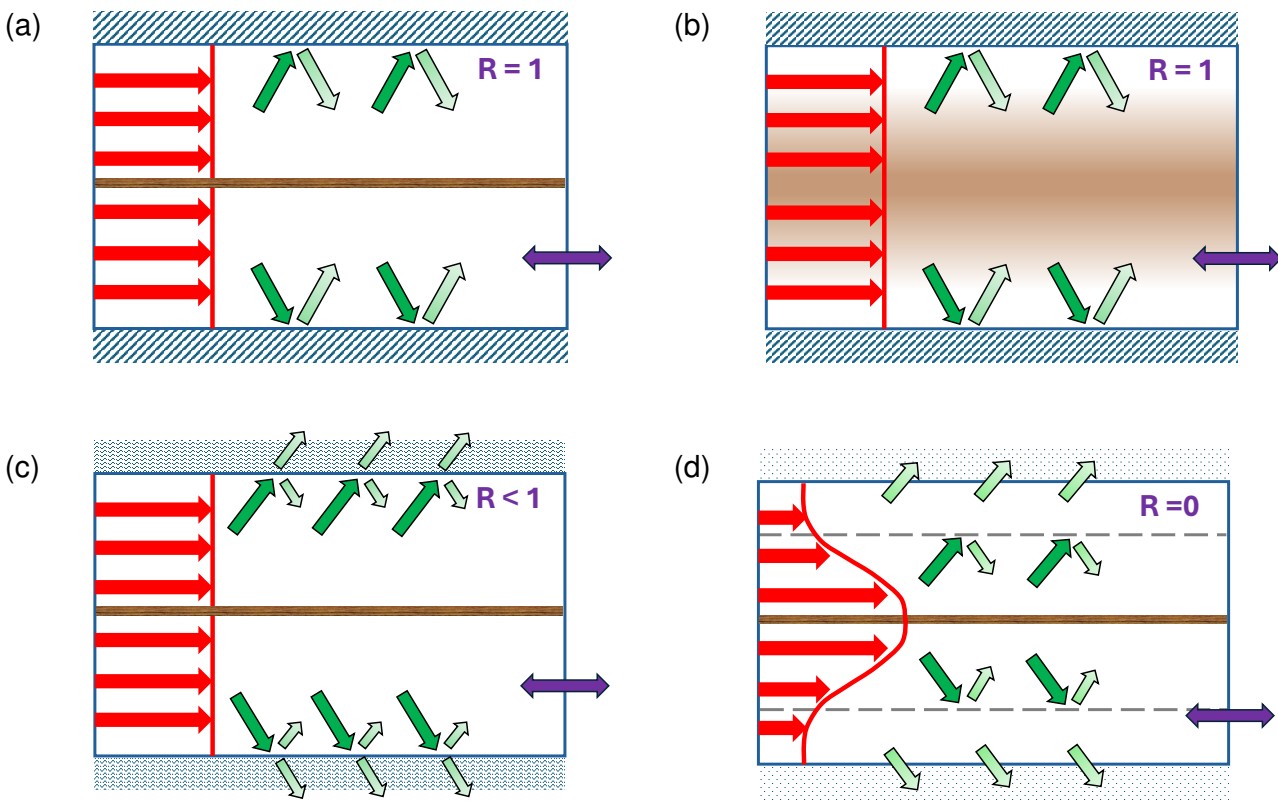

**Figure 1.** Four schematics representing the different model configurations used in this paper. The red arrows and the red line depict the basic state meridional wind $u_0(y)$, the brown color represents the forcing, the green arrows represent the wave activity flux (illustrating wave propagation, reflection, transmission, or partial reflection, respectively), the blue double arrow represents the periodic boundary conditions in the zonal direction, and the two horizontal dashed lines in (d) depict the approximate location of partial reflection at the periphery of the jet flanks.

## 3 Analytical solutions

We start with model configurations allowing analytical solutions, because these will facilitate the interpretation of the numerical solutions later in section 4. Throughout this section we assume that the basic state zonal wind is independent of latitude, i.e., $u_0 = U = const$, and this implies that the stationary wavenumber squared from (14) reduces to a constant, too, given by

$K_s^2 = \beta/U$.




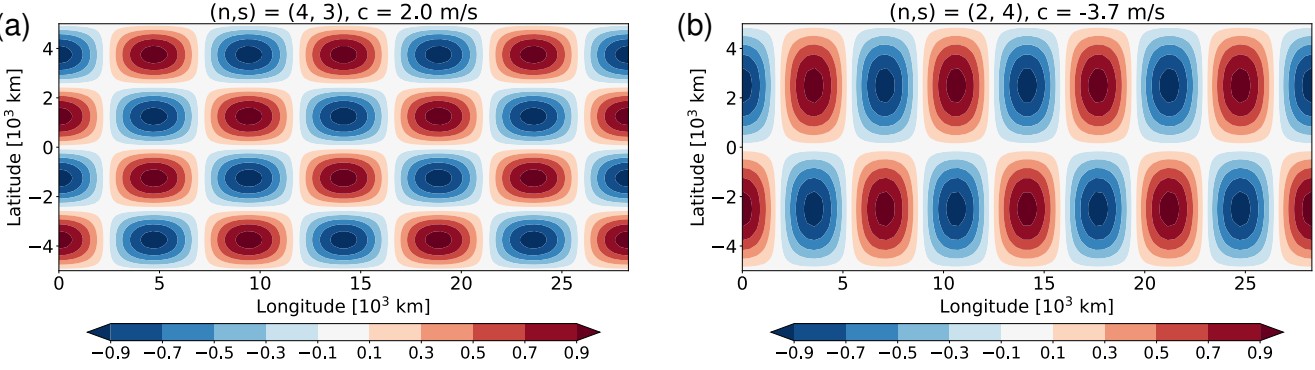

**Figure 2.** Two examples for a normal mode in a reflecting periodic channel with wavenumbers $n$ and $s$ in the meridional and zonal direction, respectively. The other parameters are $L_x$ from (1), $L_y = 10,000$ km, and $U = 10$ m s$^{-1}$. Both modes have a non-zero phase velocity $c$ as provided in the header of the respective panel.

### 3.1 Free modes and higher meridional wavenumbers

The search for free modes (or normal modes) of a linear system is motivated by the recognition that resonance occurs if the forcing projects onto a stationary free mode. Free modes are solutions of (3) with $F'$ set to zero. We restrict attention to perfectly reflecting channel walls in this subsection. The model configuration corresponds to the situation depicted in Fig. 1a, except that there is no forcing. With these assumptions, there is a discrete but infinite set of solutions, namely

$$\psi'_{n,s}(x,y,t) = \Re \psi_0 \, e^{ik(x-ct)} \sin\left[ l\left( y + \frac{L_y}{2} \right) \right] , \tag{24}$$

where the wavenumbers $k$ and $l$ are limited to discrete values given by (21) and (23), respectively, and where the phase speed $c$ satisfies the well-known dispersion relation

$$c = U - \frac{\beta}{k^2 + l^2} . \tag{25}$$

Apparently, the free modes are quantized not only in the zonal direction (due to the requirement of periodicity, non-dimensional wavenumber $s$), but also in the meridional direction (due to the finite width of the reflecting channel, non-dimensional wavenumber $n$). For illustration, we show two examples in Fig. 2, namely $\psi'_{(4,3)}$ (associated with $c = 2.0$ m s$^{-1}$) and $\psi'_{(2,4)}$ (associated with $c = -3.7$ m s$^{-1}$). The corresponding modes on the sphere are the so-called Rossby-Haurwitz waves (Haurwitz, 1940).

The discrete set of normal modes can be represented as points on the $(k,l)$-plane. This is done in Fig. 3 (blue points) for three different values of the channel width $L_y$. The horizontal rows of points in this diagram represent modes with the same value of $n$ but varying value of $s$, and the value of $n$ increases from the bottom row to the top row. Since we assume that $L_x$ is given and fixed, the distance between the horizontal rows of points depends on the value of $L_y$ according to $\Delta l = \pi/L_y$.





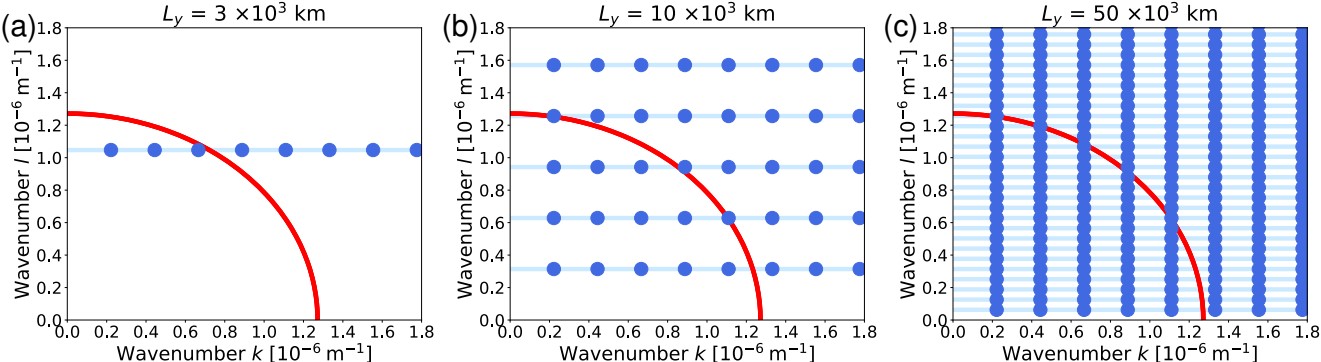

**Figure 3.** Schematic representation of the normal modes in a reflecting periodic channel with length $L_x$ from (1) and with a constant basic state wind $U = 10$ m s$^{-1}$: (a) for $L_y = 3 \times 10^3$ km, (b) for $L_y = 10 \times 10^3$ km, and (c) for $L_y = 50 \times 10^3$ km. Each blue dot represents a free modes $\psi_{n,s}$ as given in (24). The red circle with radius $\sqrt{\beta/U}$ represents the combination of wavenumbers $k$ and $l$ for which the phase speed is zero according to (25). The horizontal light-blue lines depict the hypothetical situation without the discretization constraint due to the zonal boundary condition (see explanation in the main text).

Most of the modes displayed in Fig. 3 are associated with a nonzero phase speed $c$ according to (25). The solid red line in

this plot depicts the location in wavenumer-space where $c = 0$, which is equivalent to

$$k^2 + l^2 = \frac{\beta}{U} \equiv K_s^2 \; . \tag{26}$$

All modes that lie to the bottom-left of the red circle have $c < 0$, while all modes that lie to the top-rigtht of the red circle have $c > 0$. Particularly interesting are those few modes that happen to lie on (or are very close to) the red circle, because they are (almost) stationary. These modes are associated with resonance if an appropriate stationary forcing is switched on. With

"appropriate" we mean that the forcing has a non-vanishing projection onto the respective mode.

As was mentioned before, a westerly jet can act as a waveguide, although its waveguidability is likely to be less than 1. Leakage across the jet flanks can, to some approximation, be considered as similar to damping, and even in a leaky channel one may obtain a peak in amplitude as one moves across the resonant wavenumber (Harnik and Wirth, 2025). Hence, a reflecting channel can be informative.

The important point here is that out of the three options for $L_y$ displayed in Fig. 3, only the choice in panel (a) can be taken as representative for a midlatitude jet. By contrast, the channel widths in panels (b) and (c) are much larger than the width of a typical jet. After all, the defining characteristic of a midlatitude jet is that its zonal scale is much larger than its meridional scale. As a consequence of this anisotropy, the red circle in Fig. 3a has only one intersection with the light blue line, suggesting the existence of just one resonant peak. Note that for an even smaller value of $L_y$ (not shown) there may in

fact be no intersection at all between the red circle and any of the blue lines. It transpires that for a jet-typical scenario, only



the first meridional mode ($n = 1$) is likely to contribute to resonant behavior. In other words, higher meridional modes ($n > 1$) can contribute to resonance only in channels with unrealistically large width (panels b and c). This suggests that our strategy with varying $s$ and checking the result should yield in no more than one resonant peak for any realistic jet width — and (as we will see) this is what we obtain in most cases. The location of the peak should correspond to the intersection of the red circle with the light-blue line in Fig. 3a, which in our example is at $s = 3.25$.

More formally, the condition for resonance is (26). Accounting for the quantizition (23) in the meridional direction, but considering $s$ as continuous, one obtains the following expression for the resonant wavenumber

$$s_{\text{res}} = \frac{L_x}{2\pi} \sqrt{K_s^2 - l^2} \equiv \sqrt{\hat{K}_s^2 - \left(\frac{n L_x}{2 L_y}\right)^2}, \qquad (27)$$

with $n = 1, 2, 3, \ldots$. Of course, the resulting values $s_{\text{res}}$ would be integers only by chance. Yet, to the extent that $s_{\text{res}}$ is close to an integer for one or several values of $n$, the free mode is close to resonance and one may expect that the corresponding forced solution has a large amplitude. In addition, the requirement that $s_{\text{res}}$ must not be imaginary restricts the set of allowed values of $n$ in the above relation. Given that a jet is characterized by $L_x \gg L_y$ and that typically $\hat{K}_s < 10$, it transpires that the meridional mode $n = 1$ is likely to be the only one that is associated with a physical (i.e., non-imaginary) value for $s_{\text{res}}$.

### 3.2 Charney-Eliassen forced solution

We now keep the same configuration as in the previous subsection except that we switch on forcing of the following form

$$\hat{h}(y) = h_0 \cos ly, \quad \text{with } l = \pi/L_y. \qquad (28)$$

This type of forcing was used a long time ago by Charney and Eliassen (1949) and will, hence, be referred to as Charney-Eliassen forcing. The resulting model configuration is illustrated in Fig. 1b. Note that the Charney-Eliassen forcing is less general than the delta-forcing in the sense that it contains only one specific meridional wavenumber. By contrast, the Fourier-decomposition of the delta-funcion contains all possible wavenumbers, and the solution is freer to "choose" its meridional wavenumber.

Using the Ansatz $\hat{\psi} = \psi_0 \cos ly$ (which satisfies the fully reflecting meridional boundary condition), one obtains

$$\hat{\psi}(y) = \frac{f_0 h_0}{K^2 - K_s^2} \cos ly \qquad (29)$$

and, hence,

$$\psi'(x, y) = \frac{f_0}{K^2 - K_s^2} h'(x, y), \qquad (30)$$

where $K = \sqrt{k^2 + l^2}$ denotes the total wavenumber. Apparently, the response $\psi'$ is proportional to the forcing $h'$, which is a generic property of any linear forced system. What's more interesting is the denominator on the right hand side. The latter



turns zero and, hence, the response turns infinite when the total wavenumber equals the stationary wavenumber, $K^2 = K_s^2$, or equivalently

$$k^2 + l^2 = \frac{\beta}{U} \ . \tag{31}$$

This singularity is arguably the dominant hallmark of linear resonance. Obviously, (31) is equivalent to the condition (26) for the corresponding normal mode to be stationary. In addition, the sign of $\psi'$ switches discontinuously for increasing $k$ or $s$ as one moves across the singularity, and this corresponds to a phase change by $\pi$. Such a phase change between the forcing and the response is another hallmark of linear resonance.

According to (28), the meridional wavenumber $l$ in the Charney-Eliassen configuration is determined by the meridional channel width. Correspondingly, the condition for resonance becomes

$$s \stackrel{!}{=} s_{\mathrm{res}} = \frac{L_x}{2\pi} \sqrt{\frac{\beta}{U} - \left(\frac{\pi}{L_y}\right)^2} \ . \tag{32}$$

It follows that there is either one resonant wavenumber or none, depending on whether the expression under the square root is positive or negative. Regarding the existence of free modes as illustrated in Fig. 3, one obtains only one row of blue points corresponding to $n = 1$, allowing either one or no intersection between the light blue line and the red circle. Basically, the specific meridional shape of the Charney-Eliassen forcing (28) excludes the higher meridional modes from the solution.

If one chose to include damping by adding $-\alpha q'$ (with $\alpha > 0$) to the right hand side of (3), one would obtain an additional, purely imaginary term in the denominator on the right hand side of (29) and (30). It follows that damping prevents the singularity; yet, for small enough values of $\alpha$ the functional dependence of $|\hat{\psi}|$ on $k$ or $s$ still shows a pronounced peak close to the resonant wavenumber.

### 3.3 More general forced solutions for partly reflecting channel walls

At first sight it seems that the Charney-Eliassen model configuration is not well suited to investigate Rossby wave resonance along a jet, because it makes two rather strong assumptions. First, the forcing has a very specific structure in the meridional direction, necessitating the same specific structure for the solution $\psi'$; this may be considered as dangerous, because more general forcing may project onto higher meridional modes, and it is not entirely clear at this point how this would affect the solution. Second, the Charney-Eliassen solution assumes perfectly reflecting channel walls; as was shown by Harnik and Wirth (2025), this assumption must be considered as unrealistic, because Rossby wave resonance on a jet is more akin to resonance in a channel with some leakage of wave activity across the channel walls. These two issues motivate the following modified model configuration as a better alternative: instead of Charney-Eliassen forcing we now use our delta-forcing as defined in (7), and we furthermore allow some leakage by specifying $R < 1$ at the channel walls. The model configuration for this set of experiments is illustrated in Fig. 1c.





Analytical progress can still be made by sticking to a constant wind $U$. In this case, the solution for either $y > 0$ or $y < 0$ is a superposition of plane waves like $\psi' \propto \exp i(kx + ly)$ with $k$ and $l$ satisfying (31). The coefficients must be determined through a matching condition at $y = 0$, which effectively accounts for the forcing. Using very similar methods as in Harnik and Wirth (2025), we obtain

$$\hat{\psi}(y) = \begin{cases} A\,e^{ily} + B\,e^{-ily}\,, & y \geq 0 \\ B\,e^{ily} + A\,e^{-ily}\,, & y < 0 \end{cases} \tag{33}$$

with

$$A = \frac{if_0 D}{2l\left(1 + Re^{ilL_y}\right)}\,, \tag{34}$$

$$B = \frac{-iRf_0 D\,e^{ilL_y}}{2l\left(1 + Re^{ilL_y}\right)} = -Re^{ilL_y} A\,, \tag{35}$$

and with $D$ as defined in subsection 2.2. For a fixed value of $k$, the meridional wavenumber is given by virtue of (31) as

$$l = \sqrt{\frac{\beta}{U} - k^2}\,. \tag{36}$$

Hence, for any given $U$, the value of $l$ depends on $k$ or $s$, respectively, and the solution $\hat{\psi}(y)$ depends on $k$ or $s$ in a nonlinear fashion through (34) and (35). Considering the zonal wavenumber as continuous, the relation (36) does not represent a quantization constraint for $l$, in contrast to (23)

The above solution suggests resonant behavior when the denominator in the expressions for $A$ and $B$ vanishes, i.e., when

$$l\left(1 + Re^{ilL_y}\right) = 0\,. \tag{37}$$

For fully reflecting boundaries ($R = 1$) this condition can be satisfied through the second factor on the left hand side. It requires $l$ to be real and to satisfy the following quanitzation rule

$$l \overset{!}{=} n\frac{\pi}{L_y}\,, \quad n = 1, 3, 5, \ldots \tag{38}$$

By means of (31), the above translates to a condition for $s$, namely

$$s \overset{!}{=} s_{\text{res}} \equiv \frac{L_x}{2\pi}\sqrt{\frac{\beta}{U} - \left(n\frac{\pi}{L_y}\right)^2} \tag{39}$$

with $n = 1, 3, 5, \ldots$, where the choice of admissible values for $n$ is limited through the condition that $s$ must be real. By contrast, for partial reflection ($R < 1$), there is no true singularity, although the solution still may have a pronounced peak at the values of $l$ given in (38) to the extent that $R$ is close to 1.

Comparison of (39) with the corresponding condition (32) for the Charney-Eliassen configuration indicates that the latter is a special case of the former: Charney-Eliassen only accounts for $n = 1$, while the current relation possibly allows higher meridional modes with $n > 1$.



Interestingly, condition (38) resembles, but yet is different from, the condition (23) for the existence of normal modes. More specifically, the resonant modes of our current problem correspond to only the odd meridional normal modes. The reason lies in the fact that all even modes have a node at mid-channel latitude, and this is exactly where our delta-forcing is located. It follows that the special form of our forcing in (7) allows a non-zero projection only onto the odd modes and can, therefore, trigger resonance only for this reduced set of modes.

In addition, condition (37) is satisfied when $l = 0$, and formally this corresponds to $n = 0$ in (38) and (39). In this case, the meridional flux of wave activity vanishes owing to (20), which means that wave activity is ducted in the zonal direction. Again, this should result in resonant behaviour thanks to the zonal periodicity as soon as $s$ is an integer. We will refer to this solution as the $n = 0$ meridional mode.

The two options allowing resonance are distinctly different, because the first option includes meridional wave propagation while the second does not. However, the only aspect that is relevant for resonance is the fact that wave activity is channeled in the zonal direction without leakage in the meridional direction, and this is guaranteed for both options. In the first option it is achieved through the existence of perfectly reflecting meridional boundaries, while in the second option it is achieved through the flux of wave activity being purely zonal.

We now follow our general strategy and test for resonant behavior by varying $s$ and analysing both the amplitude and the phase of the stationary solution. The result is shown in Fig. 4 for various values of $R$ and with $U$ and $L_y$ fixed at $U = 10$ m s$^{-1}$ and $L_y = 3000$ km, respectively. The amplitude of the response is measured as $\max_y |\hat{\psi}(y)|$ and its phase as the phase of $\hat{\psi}$ at the jet latitude. For the current choice of parameters, condition (39) predicts two resonant peaks, one at $s = 3.25$ corresponding to the first meridional mode $n = 1$, and one at $s = 5.73$ corresponding to $n = 0$. We first consider the behavior at $s = 3.25$. Apparently, the amplitude for $R = 1$ in Fig. 4a indicates a singularity at this value of $s$, while for smaller values of $R$ the peak gets less pronounced and vanishes completely for values $R \lesssim 0.25$. The singularity in amplitude at $s = 3.25$ in Fig. 4a is mirrored by the behavior of the phase in Fig. 4b; in particular, the phase is discontinuous at $s = 3.25$ for fully reflecting conditions ($R = 1$), giving way to a more gradual transition for $R < 1$. This general behavior in terms of amplitude and phase is very similar to the damped linear oscillator from classical mechanics; furthermore, it is consistent with Harnik and Wirth (2025), who showed that partial transmission at the channel boundaries has a similar effect on resonance as damping.

The other option for resonance implies $n = 0$, which for the current choice of parameters occurs at $s = 5.73$ according to (39). Indeed, there is a pronounced (but very narrow) peak in Fig. 4a at this location, at least for $R < 1$. Interestingly, this peak is absent for $R = 1$. Further investigation (see appendix) reveals that the limit $R \to 1$ for the $n = 0$ resonance is singular: although each of the coefficients $A$ and $B$ in (34) and (35) blow up individually for $l \to 0$, the sum of both terms on the right hand side of (33) remains finite. The singularity in amplitude is reflected by a special behavior of the phase, which shows a jump by $\pi/2$ across $s = 5.73$ for $R = 0$ and a more complicated behavior for $0 < R < 1$ (Fig. 4b). Interestingly, the value $\pi/2$ differs from the value $\pi$ that one obtains in case of the harmonic oscillator from classical mechanics. We speculate that



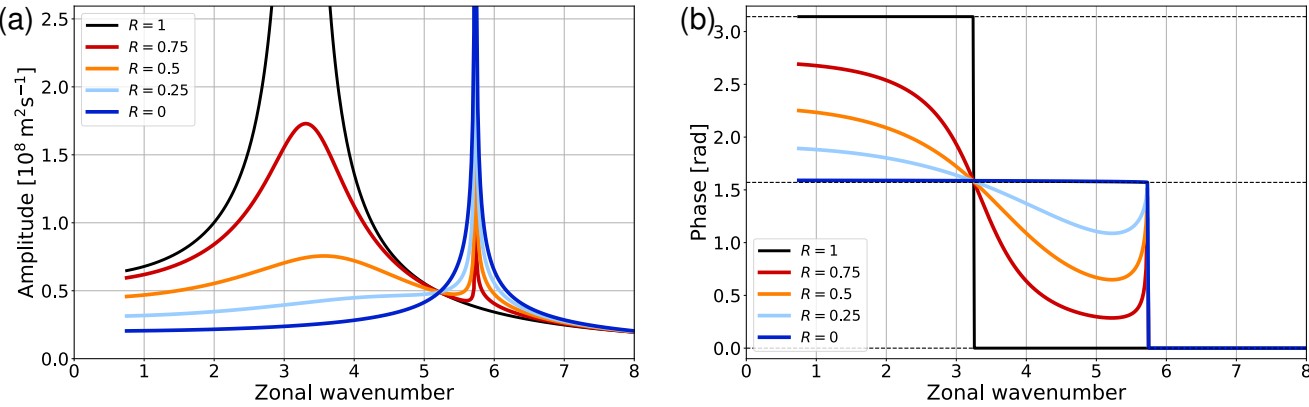

**Figure 4.** Resonant behavior of the analytic solution for delta-like forcing with $U = 10 \text{ m s}^{-1}$ and $L_y = 3000$ km. (a) Maximum value of $|\hat{\psi}|$ throughout the channel, (b) phase of $\hat{\psi}$ at $y = 0$, both ploted as a function of zonal wavenumber $s$. The different colors refer to different values of the reflection parameter $R$ (see legend); the horizontal dashed lines in (b) indicate the values $0$, $\pi/2$, and $\pi$.

this is related to the fact that there is an asymmetry as one moves across $s = 5.73$: the system supports free Rossby waves for $s < 5.73$, while it does not support free Rossby waves for $s > 5.73$.

We further illustrate the analytical solution for a number of parameter choices in Fig. 5. This figure shows the patterns of the perturbation streamfunction on the longitude-latitude plane for three different values of $R$ and three different values of $s$. First we note that fully reflecting channel boundaries (top row) always imply $\psi' = 0$ at the channel walls — by design. There

is no phase tilt with latitude, because the northward and the southward traveling wave have equal amplitude. In all other cases with $R < 1$, the solution features non-zero values at the channel walls. There is a meridional phase tilt in Fig. 5d and g close to the channel walls consistent with outward wave propagation. However, this phase tilt vanishes for $s \geq \hat{K}_s$ (second and third column), because in this case the meridional wavenumber $l$ is either zero (second column) or imaginary (third column) owing to (36); this situation is equivalent to no meridional wave propagation according to (20).

The three panels in the left column of Fig. 5 are particularly relevant for our further analysis. They represent situations which do allow meridional wave propagation. Proceeding from the top to the bottom of this column, one can identify a noteworthy transition from modal behavior for fully reflecting conditions (panel a) to plane-wave behavior for fully transparent condition (panel g). Unsurprisingly, "modal behavior" is qualitatively reminscent to the normal modes of Fig. 2. The intermediate situation for $R = 0.5$ (panel d) looks like a superposition of the two extreme cases and, thereby, contains aspects from both. We

anticipate that the intermediate case will help to understand more realistic jet profiles $u_0(y)$, because (as we will see) these are associated with partial reflection and partial transmission of wave activity at the periphery of the jet flanks.





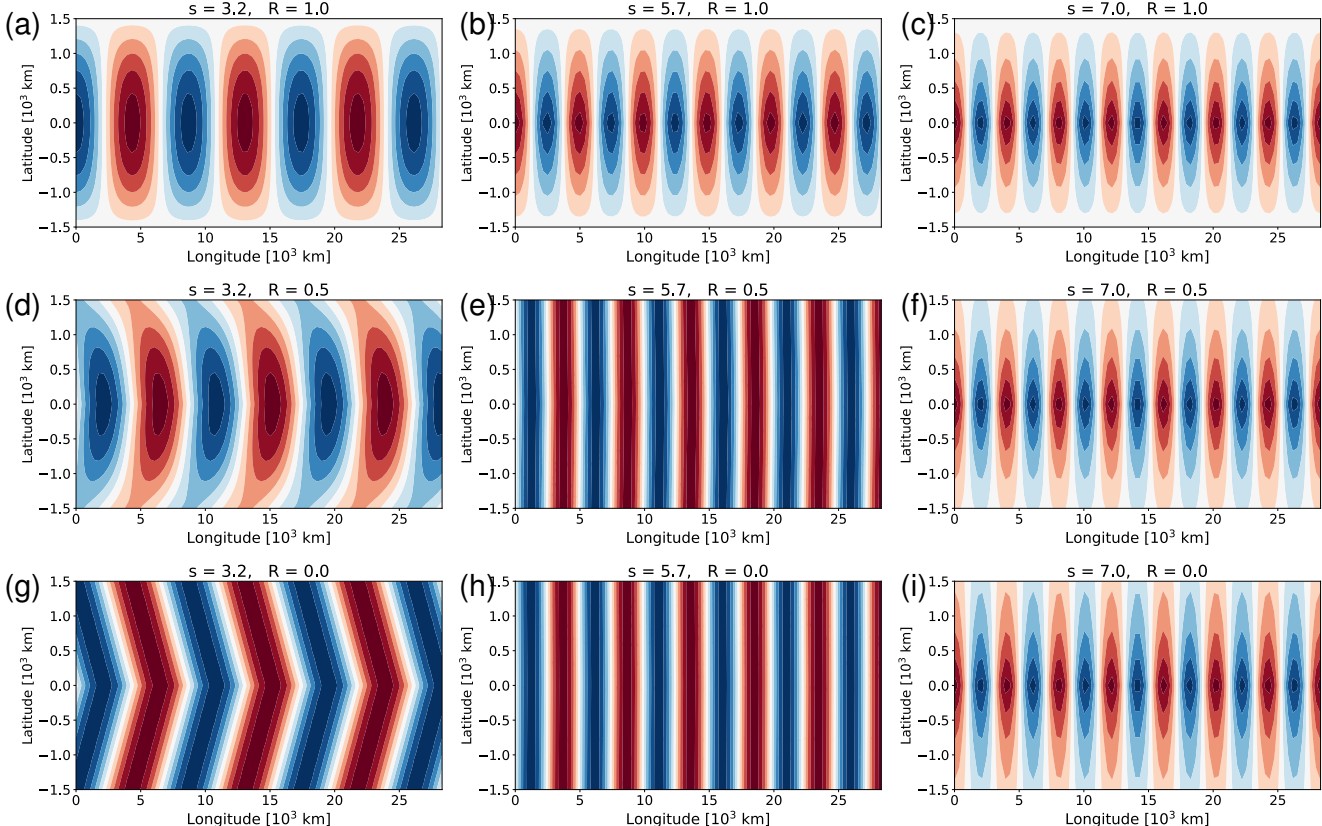

**Figure 5.** Patterns of the normalized streamfunction of the analytic solution (33) for delta-like forcing with $U = 10$ m s$^{-1}$ and $L_y = 3000$ km. The different panels represent different combinations of $s$ and $R$ (see panel caption). The range of ploted values extends from $-1$ to $+1$, with red denoting positive and blue negative values. Note that the non-normalized amplitudes in (a), (e) and (h) would be very large to the extent that $s$ is close to resonance.

We emphasize, again, the different nature of the two resonant peaks in Fig. 4a. While the $n = 1$ resonance (at $s = 3.25$) produces a sharp peak for $R \rightarrow 1$, the $n = 0$ resonance (at $s = 5.73$) produces a peak for any value of $R$ *except $R = 1$*. As mentioned earlier, for the $n = 1$ resonance the waves travel both northward and southward and keep superimposing if the

meridional wavelength happens to be just right. By contrast, for the $n = 0$ resonance the meridional flux of wave activity is zero such that wave activity that is generated at $y = 0$ cannot escape in the meridional direction and, hence, keeps accumulating within the domain. In both cases there is a physical mechanism that prevents leakage in the meridional direction and, hence, allows resonance.

We also show results for other choices of the channel width (Fig. 6). First consider $L_y = 1500$ km in Fig. 6a. Apparently, for

such a narrow channel we only obtain the peak corresponding to $n = 0$. By contrast, increasing the value of $L_y$ to 10,000 km





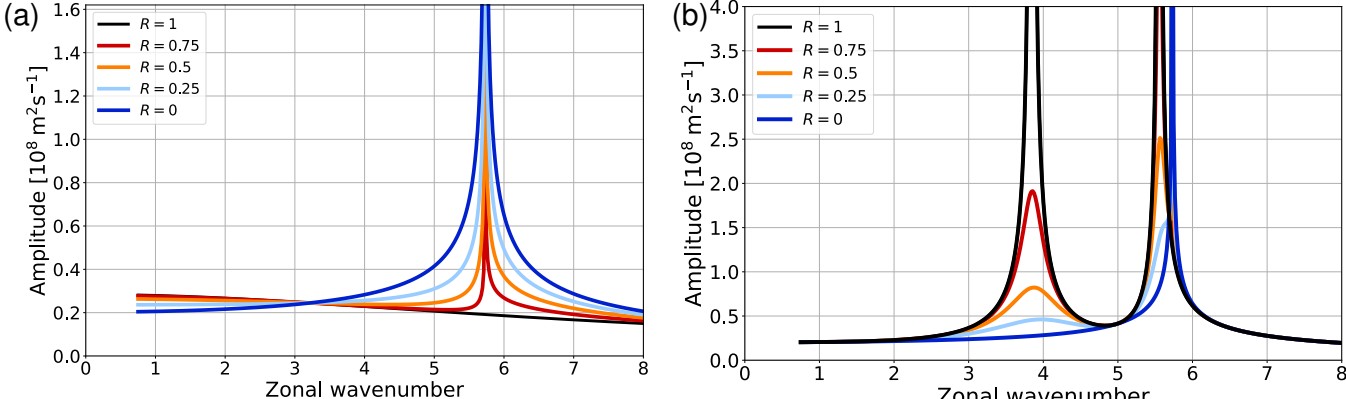

**Figure 6.** Same as Fig. 4a, except for (a) $L_y = 1500$ km, (b) $L_y = 10,000$ km.

(Fig. 6b) makes the resonant peaks for $n = 0$ (at $s = 5.73$) and $n = 1$ (at $s = 5.55$) almost coalesce, and one obtains a third peak at $s = 3.85$ corresponding to $n = 3$.

Let us briefly restrict attention to the perfectly reflecting channel (black line in Fig. 6b) and further illuminate these results by connection with the idea of resonant normal modes as illustrated in Fig 3. In the latter figure, the spacing between the horizontal

light-blue lines increases as the channel width decreases. It follows that the possibility for resonance completely vanishes when the channel becomes too narrow. On the other hand, for $L_y = 10,000$ km (Fig. 3b), there are multiple intersections between the red circle and the horizontal light-blue lines; the intersections with the first and the third light-blue line from below correspond to the peaks $n = 1$ and $n = 3$ in Fig. 6b. If one chooses an even larger (and clearly unrealistic) channel width (Fig. 3c), one obtains a very large number of intersections between the red circle and the different light-blue lines. Translated to our strategy

of diagnosing the amplitude as a function of continuous $s$, this would produce considerably more (and more densely spaced) peaks compared to those in Fig. 6b. In the limit $L_y \to \infty$, literally every real value for $s$ would be associated with resonance.

## 4  Numerical solutions

In the light of our goal to investigate jet-like basic states, we now turn attention to numerical solutions. The strategy for resonance detection will remain the same as in the previous section.





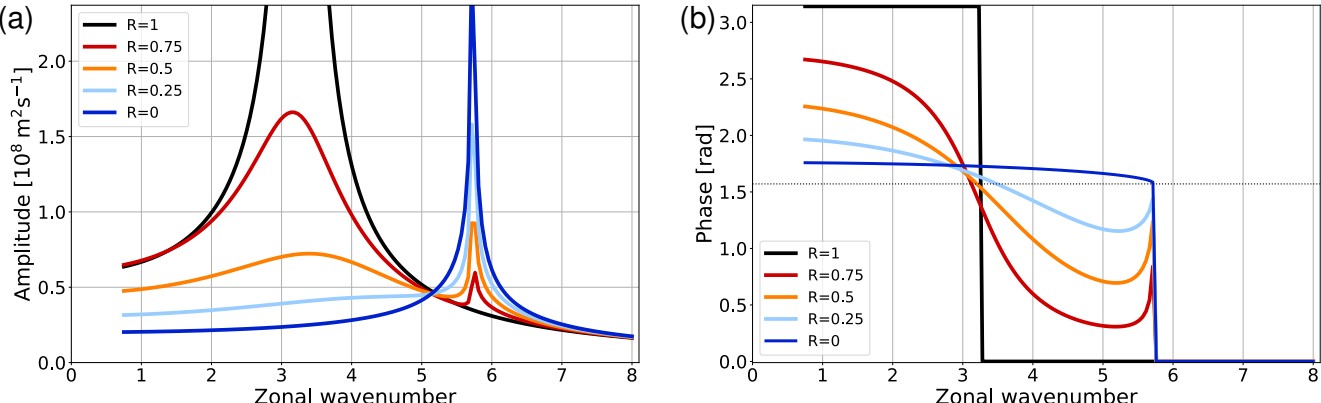

**Figure 7.** Same as Fig. 4, except for the numerical (instead of the analytical) solution.

## 4.1 Constant basic state

We start with validating our numerics by considering a constant basic state $u_0 = U = const$ and comparing the numerical solution with the corresponding analytical solution. Using $L_y = 3000$ km, we obtain the result shown in Fig. 7. Comparison with Fig. 4 indicates that the overall behavior is very similar. In particular, the location of the resonant peaks in Fig.7a is exactly where expected from (39) with $n = 0, 1, 3, 5, \ldots$, and the dependence on $R$ in both panels of Fig. 7 is qualitatively similar as in the analysical solution from Fig. 4. Admittedly, the numerical solution does not quite reproduce the exact behavior in the neighborhood of the $n = 0$ resonance. A closer examination indicates that this is presumably due to the finite meridional width of the forcing in the numerical model configuration. Overall, however, we consider the agreement between the analytical and the numerical solution as very satisfying, thus providing credibility to our numerics.

## 4.2 Transition from constant to jet-like wind profiles

We now turn to the core of our analysis and consider more realistic jet-like wind profiles $u_0(y)$. The model configuration for this set of experiments in illustrated in Fig. 1d. The latitudinal variation of $u_0(y)$ precludes a general analytical solution, but the numerical solution remains straightforward.

In contrast to earlier, we now restrict our attention to fully transparent boundary conditions at the meridional boundaries. Basically, we aim to learn whether and to what extent the flanks of the jet themselves have partly reflecting properties, and this would be confounded if we included reflection at the meridional boundaries. We posit that any amount of wave activity that manages to escape the jet region can freely propagate away towards infinity in the meridional direction. Hence, we set $R = 0$ at the meridional boundaries as a natural choice for this set of experiments. We also make sure that the entire jet is contained in our





computational domain, which means that $u_0$ transitions to a practically constant wind profile close the meridional boundaries. Incidentally, for any such wind profile our results should not depend on the meridional width of the domain owing to the fully

transparent boundaries. We checked this prediction and found that, indeed, the numerical solutions are practically independent of the choice of $L_y$.

In all considered cases, our background wind is specified to be a Gaussian westerly jet superimposed on a constant $U > 0$,

$$u_0(y) = U + (u_{\max} - U)\, e^{-y^2/(2\sigma_J^2)} \, , \tag{40}$$

with $U > 0$ and $u_{\max} \geq U$. This choice implies $u_0 > 0$ everywhere and, thus eliminates any possible singularity in (13). In

addition, it implies that barotropically unstable modes must have a positive phase velocity according to Howard's semicircle theorem (e.g. Kundu, 1990); our focus on stationary solutions thus excludes barotropically unstable modes.

In our attempt to understand the transition between a constant basic state and a jet-like basic state, we change the amplitude of the jet but keep its width constant at $\sigma_J = 500$ km. The three wind profiles used are shown in the top row of Fig. 8. The first one in Fig. 8a is a constant wind $u_0 = U = 10 \text{ m s}^{-1}$ (as before, i.e., no jet at all), the second one in Fig. 8b is a weak jet with

$u_{\max} = 16 \text{ m s}^{-1}$, and the third one in Fig. 8c is a strong jet with $u_{\max} = 30 \text{ m s}^{-1}$.

The resulting resonant behavior is presented in the third row of Fig. 8. In all three cases there is only one single peak. The interpretation of the peak in Fig. 8g is straightforward thanks to our previous analysis of the analytical solution: the sharp peak represents the $n = 0$ resonance (see the blue line in Fig. 4a), and there cannot be any further peak representing $n > 0$ because of our fully transparent meridional boundaries. Interestingly, both the location and the character of the peak changes

as one proceeds from the constant $U$ to the strong jet (Fig. 8g, h, and i). Based on the experience from the previous section, the somewhat more gradual shape of the peak in Fig 8i is reminiscent of a $n = 1$ resonance in a partly reflecting channel. This interpretation is supported by the phase behavior in the second-to-last row (panel l versus panel j). At the same time, the weak jet in the middle column of Fig. 8 represents a situation which is intermediate between the constant wind and the strong jet.

We designed a metric $Q$ which is meant to measure the "quality" or "strength" of the resonance. This is achieved by

quantifying the sharpness of the peak in the functional dependence of $f(s) = (\max_y |\hat{\psi}|)(s)$ as shown in the third row of Fig. 8. We first determine $s_{\mathrm{res}}$ as the wavenumber at which the function $f(s)$ maximizes, and then we define

$$Q = \frac{2f(s_{\mathrm{res}})}{f(s_{\mathrm{res}} - 1) + f(s_{\mathrm{res}} + 1)} - 1 \, . \tag{41}$$

The diagnostic is designed such that $Q = 0$ if $f(s)$ is a constant function, and $Q$ increases to the extent that the maximum represents an increasingly narrow peak. The value $Q = 1$ is reached when (for a symmetric peak) the maximum value is twice

as large as the ambient values at a distance $\Delta s = \pm 1$. This value ($Q = 1$) can be taken as a meaningful threshold for the occurrence of resonance in meteorological applications. The $Q$-values for the three basic states in Fig. 8 are provided in the respective panels in the third row. Apparently, the $n = 0$ peak in the left column is very sharp resulting in a very high value





**Figure 8.** (Figure caption on following page.)




**Figure 8.** Numerical analysis for different basic states with increasing jet-like meridional variation of $u_0(y)$ in a zonally periodic domain with fully transparent meridional boundaries. The top row shows the zonal wind $u_0(y)$; the second row shows the stationary wavenumber $\hat{K}_s(y)$, where the negative values (shading) represent minus the imaginary part of $\hat{K}_s$ ; the third shows the maximum amplitude $\max_y |\hat{\psi}|$ as a function of $s$; the fourth row shows the phase of $\hat{\psi}$ at $y=0$ as a function of $s$; and the bottom row shows the pattern of the normalized perturbation streamfunction at the value of $s$ that corresponds to the peak in the amplitude plot (third row). The plot conventions in the bottom row are the same as in Fig. 5. Key characteristics $s_{\mathrm{res}}$ and $Q$ of the solution are provided in the panels of the middle row.

$Q \gg 1$. By contrast, the weak jet is associated with a value $Q < 1$, while the strong jet has $Q > 1$. The latter behavior is consistent with the conventional wisdom that stronger jets are better waveguides, implying a stronger tendency for resonant behavior.

Can we "understand" the transition between the solutions shown in the three columns in Fig. 8? The $n = 0$ resonance visible in the left column gets weaker to the extent that the wind profile has an increasing amount of latitudinal variation, and this happens presumably for two reasons. First, the value of $s_{\mathrm{res}}$ in (39) for $n = 0$ turns less well defined to the extent that the basic state wind is not a constant any longer. At the same time, the latitudinal variation of $u_0$ prevents a unique value of $l$ in (36). As a consequence, the condition $l = 0$ can be satisfied only at one specific latitude rather than within a whole range of latitudes. Second, we posit that the flanks of a jet are associated with at least partial reflection $R > 0$, with increasing values of $R$ for increasing jet strengths (Manola et al., 2013; Wirth, 2020; Harnik and Wirth, 2025). Fig. 6a suggests that the width of the $n = 0$ resonant peak decreases as the value of $R$ increases, and this means that the $n = 0$ resonant peak gets less dominant. At the same time, as one starts to build a jet, this jet is associated with an increasing amount of reflection, and one starts to obtain an $n = 1$ resonant peak; the strength of this peak should increase for stronger values of $R$ (Fig. 4a) and, hence, for stronger jets. This interpretation is supported by the fact that the peak in panel (i) is rather wide, while the peak in panel (g) is very sharp — consistent with the different shapes of the peaks for $n > 0$ versus $n = 0$ in our analytical solutions from the previous section. In summary, we suggest that there is a gradual "blend-out" of the $n = 0$ peak and a gradual "blend-in" of an $n = 1$ peak as one proceeds from the constant wind to the strong jet. Apparently, the solution for the weak jet in the middle column of Fig. 8 lies somewhere in between these two extremes.

The interpretation offered above is consistent with the patterns of the perturbation streamfunction for the three solutions (bottom row in Fig. 8). The constant basic state (panel m) shows — by design — the behavior from the corresponding analytical solution (Fig. 5h). The other two scenarios (panels n and o) show an increasing amount of confinement of wave activity to the jet-region, with outgoing plane waves beyond the jet region. In particular, the pattern in panel (o) is reminiscent of the pattern of the analytical solution for a partly reflecting (or partly leaking) channel from Fig. 5d.





### 4.3 Interpretation in terms of partial reflection at the periphery of the jet flanks

We now aim to corroborate the interpretation of the resonant behavior for the strong jet case (right column of Fig. 8) in terms of approximate partial reflection at an internal interface at the periphery of the jet flanks. As mentioned earlier in connection with (17), the two key variables in this equation are $\hat{K}_s$ and $s$, determining whether the local character of the solution is wavelike

or exponential. Let us apply this diagnostic concept with the help of Fig. 8e and f. For those values of $s^2$ that lie between the relative maximum of $\hat{K}_s^2$ at the jet core and the relative minimum of $\hat{K}_s^2$ at the jet's flank, the character of the solution switches from wavelike in the jet core to exponential at the jet periphery. The stronger the jet, the larger is the corresponding range of wavenumbers. We hypothesize that the "exponential regions" at the jet's periphery act as partial reflectors and, hence, generate a certain amount of waveguidability.

Let us shed more light on this hypothesis. We assume that for both subdomains $y > 0$ and $y < 0$ the total perturbation streamfunction consists of two parts: the transmitted part $\psi'_{\text{trans}}$ which is able to leave the domain, and the remainder $\psi'_{\text{refl}}$ which participates in the reflection, i.e.,

$$\psi' = \psi'_{\text{refl}} + \psi'_{\text{trans}} \ . \tag{42}$$

Close to northern meridional boundary, the transmitted part has the form of an outgoing plane wave, i.e.,

$$\psi'_{\text{trans}} = \hat{\psi}_{\text{trans}} \, e^{i(kx+ly)} \ . \tag{43}$$

This part of the solution is obtained by, first, computing $l$ from (36) with $u_0(\pm L_y/2)$ substituted for $U$, and then inferring $\hat{\psi}_{\text{trans}}$ from the known full solution at the domain boundaries. The reflected part $\psi'_{\text{refl}}$ is then obtained from (42). This procedure is carried through separately for the two subdomains $y > 0$ and $y < 0$.

The pattern of the resulting $\psi'_{\text{refl}}$ is shown in Fig. 9 for the two jet-like wind profiles from the middle and right column in 460 Fig. 8. Apparently, there is very little phase tilt with latitude in $\psi'_{\text{refl}}$. The behavior is consistent with the notion that this part of the solution is associated with reflection at a latitude somewhere between the middle of the domain and the domain boundaries, resulting in a modal pattern of streamfunction (cf. Fig. 2). In fact, one may associate the reflected part $\psi'_{\text{refl}}$ with an effective channel width by estimating the latitudes at which the amplitude goes to zero. In Fig. 9b, this happens at $y \approx \pm 1200$ km; thus the effective channel width associated with this jet is $L_y \approx 2400$ km. Note that this value is considerably larger than the 465 meridional extent of the "jet cavity" where $\hat{K}_s^2 \geq s_{\text{res}}^2$, and that this is expected according to what we mentioned in the text behind (17). Furthermore, the stronger jet (panel b) is associated with a considerably stronger $\psi'_{\text{refl}}$ than the weaker jet (panel a), and this is consistent with the accepted wisdom that stronger jets are a better waveguides.

To the extent that our strong jet produces partial reflection and a mode-like behavior in the core of the jet, we should be able to relate this interpretation to the analytical solution in a reflecting channel with constant $U$. More specifically, we aim to 470 predict the resonant wavenumber from (27). Using the above estimate of the effective channel width $L_y \approx 2400$ km and the



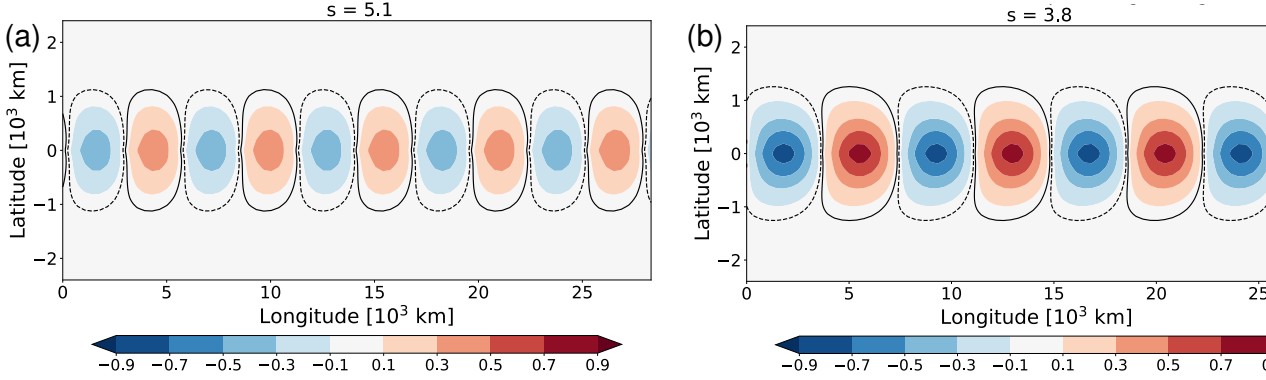

**Figure 9.** Normalized streamfunction from the two jet-like basic states of Fig. 8, but with the outgoing plane-wave parts of the solution subtracted. (a) Weak jet from the middle column of Fig. 8, (b) strong jet from the right column of Fig. 8. The solid and dashed black contours depict the values $\pm 0.03$. The numerical factor used for normalization is the same as in Fig. 8n and o, respectively.

value of $\hat{K}_s \approx 7$ close to the jet core (Fig. 8f), we obtain a single resonant wavenumber for $n = 1$ at $s_{\mathrm{res}} \approx 3.8$, which happens to be identical to the diagnosed value of $s_{\mathrm{res}} = 3.8$ in Fig. 8i. To be sure, the estimated value of $s_{\mathrm{res}}$ sensitively depends on the chosen values for $L_y$ and $\hat{K}_s$, none of which are well-defined in the jet-scenario. Yet, we consider this result as a "sanity-check", adding confidence to our interpretation in terms of partial reflection at a latitude close to the periphery of the jet's
flanks.

It is also illuminating to shift the orography in the meridional direction away from the jet. More specifically, we extend the southern part of the domain to $y = -5000$ km and shift the pseudo-orography to $y = 3000$ km. The forcing thus lies outside of the jet, in a region with constant wind $u_0 = 10\,\mathrm{m\,s^{-1}}$. For any $s < 5.7$ we expect that locally there must be two plane waves emanating from the new forcing location similar as in Fig. 5g. However, the wave that travels northward is going
to encounter the jet. Based on the earlier results from this subsection, one may expect multiple reflection between internal interfaces located at the periphery of the jet flanks such that only part of the wave activity is able to escape the jet region and leave the domain through the northern boundary. If the zonal wavenumber of the forcing happens to be $s = 3.8$, these multiply reflected waves should interfere constructively resulting in increased wave amplitudes at the jet latitude. Indeed, this is exactly what our numerical solution shows (Fig. 10). Apparently, the presence of the jet is able to generate a modal structure
within the jet region; this effectively channels wave activity in the zonal direction and, thus, leads to increased wave amplitudes owing to repeated superposition thanks to the periodic boundaries. Note that the magnitude of this maximum response in the jet core turns out to be considerably smaller (viz., only one third) compared to the value obtained in Fig. 8i; however, this is qualitatively to be expected, because the shifted forcing has a smaller projection on the resonant mode (cf. Fig. 8o).



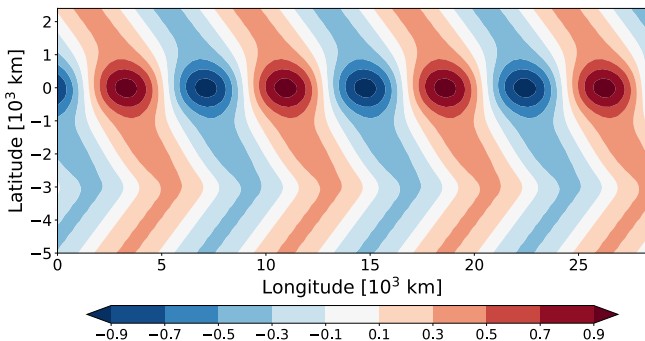

**Figure 10.** Normalized streamfunction for the strong jet similar as in the right column of Fig. 8, except that the latitude of the forcing was shifted southward by 3000 km. The zonal wavenumber was chosen to be $s = 3.8$ and, thus, to correspond to the resonant wavenumber for this jet (see Fig. 8i.)

## 4.4 Varying the jet width

Finally, we consider the resonant behavior for jet-like wind profiles as the width $\sigma_J$ of the jet varies while its amplitude $u_{\max}$ is kept constant (Fig. 11). In this set of experiments, both the domain width $L_y$ and the value of $\tilde{D}$ in (8) are varied in the same proportion as $\sigma_J$; this device guarantees that the meridional extent of the orography is always considerably smaller than the jet width and, at the same time, the orography is numerically well resolved in each experiment.

There is a wide range of behavior of the refractive index across the three experiments depicted in Fig. 11. The function 495 $\hat{K}_s(y)$ shows a relative maximum at the jet latitude for the narrow and the intermediate jet, but a relative *minimum* for the wide jet (note that the our wide jet is unrealistic in the sense that it would not fit into the midlatitudes on planet Earth). The relative minimum in this case can be explained by noting that in the wide-jet limit the second derivative of the wind profile can be neglected in (4); the stationary wavenumber squared becomes

$$K_s^2(y) \approx \frac{\beta}{u_0(y)} \, , \tag{44}$$

from which one obtains a local *minimum* of $\hat{K}_s(y)$ at the jet core. By contrast, in the narrow jet limit the stationary wavenumber squared scales like

$$K_s^2 \approx -\frac{1}{u_0}\frac{d^2 u_0}{dy^2} \sim \frac{1}{\Delta_{\mathrm{jet}}^2} \, . \tag{45}$$

In this case one expects a sharp relative maximum of $\hat{K}_s(y)$ at the jet latitude that scales like $\Delta_{\mathrm{jet}}^{-1}$.

In case of the wide jet, $\hat{K}_s = 3.5$ in the jet core (Fig. 11f), and this value is identical to the location of the resonance 505 ($s_{\mathrm{res}} = 3.5$, Fig. 11i). In other words, $s_{\mathrm{res}} = \hat{K}_s(0)$, and this implies $n = 0$ according to (27). Indeed, this result is broadly consistent with the qualitative behavior of the phase (fourth row), which suggests a transition from an $n = 1$ resonance for the





**Figure 11.** Numerical analysis in a domain with fully transparent meridional boundaries for different basic state profiles with a Gaussian jet of varying width: $\sigma_J = 250$ km (left column), $\sigma_J = 750$ km (middle column), and $\sigma_J = 2500$ km (right column). Plot conventions are like in Fig. 8.




intermediate jet to an $n = 0$ resonance for the wide jet. Moreover, as we increase the width of the jet even further (not shown), the quality-measure $Q$ of the resonant peak increases substantially, consistent with the very sharp peak of our analytical solution in the constant wind case with $R = 0$ (blue line in Fig. 4). As we will argue below, the scenario of our wide jet is unlikely to

be relevant in practice, but the consistent interpretation is nevertheless satisfying.

By contrast, the solution for the intermediate jet in the middle column of Fig. 11 has the flavor of a first ($n = 1$) meridional mode. As discussed in the previous section, this mode is established through reflection of wave activity at the periphery of the jet flanks. The fact that the resonant peak is located at a very similar wavenumber for the intermediate and for the wide jet (panels h and i) must be considered as fortuitous: apparently, the change of $\hat{K}_s(y)$ and the change of the "effective channel

width" nearly compensate each other.

Let us finally turn to the narrow jet in the left column of Fig. 11. Panel g shows a single resonant peak at roughly the same wavenumber as for the intermediate jet (panel h), but the sharpness (i.e., the $Q$-value) of the peak is considerably lower. The weakness of the resonance appears plausible in view of Fig. 3: given that the width of an "equivalent reflecting channel" would be only about 1000 km, this should actually prevent the existence of a stationary normal mode and, hence, the occurrence of a

well-defined resonant peak. Of course, this argument has to be taken with a grain of salt, since the narrow jet cannot possibly be modelled through a constant wind in a quantitative manner.

## 5    Summary and conclusions

In the current paper, we investigated Rossby wave resonance for idealized jets on a beta-plane and proposed a novel method for diagnostic purposes. As a framework we use the barotropic model, linearized about a zonally symmetric basic state. The

system is subject to forcing with a fixed zonal wavenumber $s$, with a narrow extent in the meridional direction, and with a maximum at the jet latitude. For our jet experiments we use fully transparent meridional boundaries corresponding to a radiation condition. The stationary solution is obtained through straightforward numerical methods. We then systematically vary the value of $s$ while keeping the amplitude of the forcing fixed and analyse how the solution changes as a function of $s$. Whenever the solution features a pronounced peak in amplitude at some value of $s$ and the phase crosses the value $\pi/2$ with

a steep slope, we associate the underlying basic state with a considerable potential to produce resonance. Finally we quantify the strength of resonance by the sharpness of the peak in amplitude. In addition to the numerical solutions for jet profiles, we considered a number of analytical solutions for special cases, helping us to interpret our numerical solutions.

Our main results are as follows:





– We did obtain weakly resonant behavior for various model configurations and basic states that can be considered as representative for a circumglobal midlatitude jet. It follows that the waveguiding properties of such jets are strong enough to allow a weak form of resonance to the extent that the waves are not subject to other forms of damping.

– Even a good zonal waveguide in the form of a strong jet is not associated with a true singularity in wave amplitude at the resonant wavenumber, i.e., its amplitude remains finite instead of going to infinity. This behavior is consistent with the findings of Harnik and Wirth (2025) who showed that a jet behaves qualitatively like a leaky channel. It follows that the question of Rossby wave resonance should not be framed as a binary question (resonance: yes or no?); rather it is more appropriate to talk about the "strength of resonance" or the "propensity to resonance". The situation is similar as with the concept of a waveguide, which more appropriately is framed in terms of a "waveguidability" (Manola et al., 2013; Wirth, 2020).

– For all jets with Earth-like dimensions we obtained one single resonant peak, occurring at one specific wavenumber $s_{\text{res}}$. In most cases, the resulting streamfunction was characterized by enhanced values in the jet region and outgoing plane wave behavior away from the jet region. These solutions could be interpreted as an approximate realization of the first meridional mode arising from partial reflection from a region at the periphery of the jet.

– The absence of higher meridional modes is fundamentally related to the anisotropy of a jet, i.e., to the fact that its zonal scale is much larger than its meridional scale. It can be understood with reference to resonance in a narrow reflecting channel on a constant basic state wind. Here, the condition for resonance represents a constraint which only allows specific combinations of the zonal and the meridional wavenumber. In this simple model configuration, both wavenumbers are quantized, and the narrowness of the channel implies that only the first meridional mode can be associated with resonance (if at all) for realistic scales.

– In the light of the previous two items, it appears that the notion of internal interfaces with partial reflection is a better approximation to describe the meridional propagation of Rossby waves along a jet than the framework of gradual variation of the basic state.

– Even for the extreme case of a constant basic state wind (or an unrealistically wide jet) with fully transparent meridional boundaries, our numerical solution showed a sharp resonant peak. This peak is based on the fact that a constant zonal wind allows a plane wave solution with purely zonal wave activity propagation, thus representing a perfect zonal waveguide. However, this scenario is unlikely to be important in practice, because a latitude-independent wind profile is a poor representation of typical midlatitude conditions. Moreover, this effect does probably not have a straightforward equivalent in spherical geometry.

– As we systematically varied either jet amplitude or jet width, we could observe a transition between the two types of resonance mentioned above. Given that the resonant peak associated with purely zonal plane-wave propagation is unlikely to be relevant in practice, these "transitions" have to be interpreted with care.



– Despite its strong idealizations, the Charney-Eliassen model turns out to be a surprisingly good guide to estimate resonant behavior. Assuming that the channel width in this model is chosen to correspond to the meridional scale of the jet, the Charney-Eliassen solution yields either one resonant peak, or none (when the jet width is very small). These predictions correspond well to the results from our numerical solutions, which show generally one resonant peak, but for which the
570       strength $Q$ of the resonance becomes very small for very narrow jets.

Obviously, this study comes with a few caveats and limitations that need to be kept in mind.

– We restricted our analysis to inviscid Rossby waves on a circumglobal jet, representing favorable conditions for resonance. In reality, Rossby waves are subject to various forms of damping (in addition to damping through jet leakiness), implying that the resonant response would be considerably weaker than documented in our analysis (cf. Harnik and
575       Wirth, 2025). Similarly, we expect the resonant behavior to be considerably less pronounced than one might conclude from our analysis to the extent that a jet is not circumglobal. This is because resonance of the type that we considered requires continued superposition of waves that travel around the earth multiple times along a constant latitude circle.

– We used Cartesian geometry, which implies a symmetry between the northward and the southward direction. By contrast, in spherical geometry there is a natural tendency for equatorward wave propagation, which has no equivalent in Cartesian
580       geometry.

– We only considered stationary solutions. This means that we did not address the question how long it takes until the steady state has been established. To the extent that the stationary solution is characterized by a sharp resonant peak, a sudden change in the basic state may lead to a substantial change in the wave amplitude during the subsequent transient adjustment, and it would be important to further investigate such a transient scenario.

– We focused on the jet region and assumed that any wave activity that leaves the jet region propagates further away and is not able to return towards the jet. In other words, we neglected reflection from any region located outside of the jet region. In particular, we did not address the question of whether and to what extent a critical level located in the subtropics may effectively act as a reflecting surface (Held, 1983).

In our future work we plan to add realism to the suggested method by moving to spherical geometry, including wave
damping, and using basic states derived from observations.

Overall we conclude that Rossby waves on a midlatitude jet may be subject to a weak form of resonance, provided that the jet is truly circumglobal and the wave damping is small. Given that the zonal scale of a jet is much larger than its meridional scale, one may expect resonance at no more than one zonal wavenumber $s_\mathrm{res}$. This single resonant peak is associated with the first meridional mode, which is established through partial reflection of wave activity at the periphery of the jet flanks.





**Appendix A: Singular limit of the analytical solution for $l \to 0$.**

For any $R < 1$, the term $\left(1 + R e^{ilL_y}\right)$ in the denominator of (34) and (35) is nonzero, and both $A$ and $B$ blow up individually in the limit $l \to 0$. In addition, the two coefficients satisfy

$$|B| = R|A| \,, \tag{A1}$$

which implies that there cannot be a systematic cancellation between the two terms on the right hand side of (33). It follows
that the solution for $R < 0$ tends to infinity in the limit $l \to 0$.

The situation is distinctly different for $R = 1$. In this case, the coefficients can be rewritten as

$$A = \frac{i f_0 D\, e^{-ilL}}{4l \cos(lL)} \,, \tag{A2}$$

$$B = \frac{-i f_0 D\, e^{ilL}}{4l \cos(lL)} \,, \tag{A3}$$

with $L = L_y/2$. This implies $|A| = |B|$, which opens the possibility that the two terms on the right hand side of (33) cancel
each other. Indeed, substitution of these expressions for $A$ and $B$ into (33) yields

$$
\begin{aligned}
\hat{\psi}(y) &= \frac{i f_0 D}{4l \cos(lL)} \left[ e^{-il(L-y)} - e^{il(L-y)} \right] \\
&= \frac{f_0 D}{2l \cos(lL)} \sin l(L-y)
\end{aligned}
\tag{A4}
$$

for $y \geq 0$ (and a similar expression for $y < 0$). Thus, in the limit $l \to 0$, one obtains

$$
\hat{\psi}(y) \to
\begin{cases}
(f_0/2)D\,(L-y)\,, & y \geq 0\,, \\
(f_0/2)D\,(y+L)\,, & y < 0\,.
\end{cases}
\tag{A5}
$$

The latter expression does not contain the parameter $l$ any longer, so it is well-behaved and remains finite in the limit $l \to 0$.
Note that this solution satisfies the correct boundary condition $\psi = 0$ at $y = \pm L_y/2$ (see Fig. 5b).

*Author contributions.* The first author designed the study, carried out the model experiments, and wrote the paper. The second author made essential contributions through repeated discussions about the scientific content of the paper and through comments on an earlier version.

*Competing interests.* Nili Harik is member of the editorial board of the journal Weather and Climate Dynamics.



*Acknowledgements.*   We acknowledge very helpful comments by Dr. Michael Riemer on an earlier version of this paper.



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
