# Peer review of "Rossby wave resonance for idealized jets on a beta-plane: towards a better understanding of the meridional wave structure"

_EGUsphere, 2025_

## Author Comment (AC1)

**Authors Comment:**
**Response to the editor's and the reviewers' remarks**

Below, the editor's and the reviewers' remarks are repeated (typewriter font), each followed by our reply and explanation (normal font) as to how we intend to account for their comments.

**Editor's remarks**

```
Dear Authors,

We have now received two reviews of your manuscript.  In light of these and based
on my own assessment, I would like to invite you to prepare your responses to
both reviews.  Please address in particular the comments made by reviewer no.
2 regarding the degree of novelty.

From a scientific perspective, I would like you to consider the following for
your discussion.  As you increase the sharpness of the jet, there is a possibility
of it becoming barotropically unstable.  In such cases, seeking stationary solutions
may not be meaningful.

Best wishes,

Sebastian Schemm
```

We thank the editor, Sebastian Schemm, for handling our paper and we are grateful for his remarks.

- Regarding the novelty of this work, we explicitly address this issue in our reply to reviewer 2.

- Regarding the possibility of barotropic instability: Yes, you are right, our strong jets are barotropically unstable according to Rayleight's inflection point theorem. We want to address your remark by making three points.

  – In a more realistic framework aiming to represent the Earth's troposphere, one is more likely to obtain baroclinic instability rather than barotropic instability. Obviously, this is one of the fundamental limitations of the barotropic model: it is unable to represent baroclinic instability. Real jets in the real atmosphere occur in the presence of large gradients of the background Ertel potential vorticity (PV); however, this gradient is, to a large extent, due to a quasi-horizontal gradient in static stability rather than in relative vorticity. Again, since the barotropic model "does not know" anything about static stability, it cannot possibly represent this feature of the real atmosphere. Despite this caveat, the linear barotropic model has frequently been used to study Rossby wave resonance, and in the current paper we follow this line of investigation; our focus is on an improved understanding of this model framwork, especially as far as the meridional dimension is concerned.

  – Previous investigations about Rossby wave resonance in observed episodes have usually focused on stationary waves, since stationary waves are more likely to be associated with extreme weather than traveling waves (Fragkoulidis and Wirth,

2020). This focus was achieved through time averaging, like, e.g., by analysing monthly means (Petoukhov et al., 2013) or by preprocessing with a 15-day running mean (Kornhuber et al., 2017). Therefore, we think that it makes sense to focus in our paper on stationary waves and their possible resonant amplification, and ignore the fact that the used model is unable to represent other phenomena (like baroclinic instability).

– Rossby wave resonance is typically discussed in the context of stationary waves, i.e., the resulting large-amplitude waves are produced stationary forcing. By contrast, baroclinic instability does not require any forcing and the wave growth occurs through purely internal dynamics. These two issue are conceptually well separated, and formally they can be distinguished by their zonal phase speed (close to zero in the case of quasi-stationary waves, versus eastward in the case of baroclinic waves). In a linear framework, different wave modes with different phase speeds can be dealt with independently of each other. The focus on quasi-stationary waves (see previous item) is, therefore, mathematically well-posed.

We will address the issue with the potential for barotropic instability in a future version of this paper.

**Reviewers' comments**

We thank the reviewers for carefully reading our manuscript and making insightful comments. Both reviewers mentioned that they enjoyed reading/reviewing the paper, and we are grateful for this particular feedback.

**Reviewer 1**

**Reply to his minor comments**

1. Typo will be corrected.

2. In the Introduction, I would really appreciate the authors including a more detailed comparison with several relevant papers on resonance, such as Charney and DeVore, Tung and QRA, to better position this work in the existing literature.

   We agree that it is a good idea to provide more context regarding the previous literature about resonance. Accordingly, we plan to add some text. Incidentally, these added pieces of text will also help to address the second reviewer's remark regarding the novelty of our work.

3. From equations 5 to 8, it seems that h hat is not dimensionless if D=500 km, in equation 7, which appears inconsistent with the units in equation 8. I would suggest adjusting the delta function to have the dimension of $(length)^{-1}$. Additionally, a brief explanation of the forcing expression of pseudo topography could be helpful for readers less familiar with this formulation.

   The parameter $D$ in (7) has dimensions of m, and the delta-function has dimensions of m$^{-1}$. Taken together, this renders $\hat{h}$ and, hence, $h'$ dimensionless. We will add a sentence to make this point explicit.

Also, we will modify our terminology and now talk about "pseudo-orography" instead of simply "orography", since the latter would be somewhat strange in a barotropic model.

4. In Figure 1, the annotation for \meridional wind" need to be corrected to \zonal wind."

Right, of course! Thank you for spotting this error. We will correct the figure caption accordingly.

5. I suggest adding more explanation regarding the psi value in equation 29, for example, by showing how it is derived by substituting back into equation 17.

That's a good idea. We will enhance our text in order account for the reviewer's comment.

6. When discussing the physical interpretation or the limitations of the n=0 solution, either in the Results or Discussion sections, it may be helpful to explicitly note that the presence of singularities indicates that, beyond analyzing wavenumber, amplitude, and phase, the linear equations may no longer be valid for the solution.

We are not sure whether we fully understand the reviewer's comment. To be sure, in the presence of resonance, wave amplitudes will grow linearly with time, and at some point linearization breaks down as a useful approximation. In the real world, nonlinear effects (amongst others) start to play a role, and this implies that the results from linear theory must be taken with a grain of salt. However, this does not reduce the value of linear analysis to zero, and linear theory has been applied to diagnose Rossby wave resonance often times in the past. In fact, a thorough understanding of linear resonance may prove useful to understand nonlinear effects including multiple equilibria.

In response, we are planning to add new text to the concluding section in order to explicitly address this issue. This new text will point to the obvious limitations of linear theory and clarify our primary aim, which is to better understand recent papers that do use the linear barotropic model.

7. Typo will be corrected.

8. Typo will be corrected.

9. However, it seems that a zonal resonance solution (n=0) is not necessarily impossible in realistic conditions. For example, in line 557, the jet corresponding to the n=0 solution is described as \unrealistically wide," yet Figure 6 still shows a solution with Ly=1500km, though its amplitude is weaker. Some clarification of this point could strengthen the discussion.

You are right: for a constant basic state wind with partly or fully transparent boundaries one obtains the $n = 0$ resonance even for a rather narrow domain (as exemplified in our Fig. 6). However, this sharp peak is only generated from that part of the solution, which is able to penetrate the boundaries; this part of the solution is a plane wave with purely zonal propagation, which implies that the effective width of the channel is infinite (for this part of the solution). Therefore, its mechanism is fundamentally different from the mechanism for the higher meridional modes $n = 1, 2, \ldots$, which require meridional propagation and (partial) reflection in the meridional direction.

We plan to enhance the text in this section in order to better indicate what we mean and to make a connection with what we said earlier in section 3.3. At the same time, we choose to keep the discussion short, because this peak does not have an analog in spherical geometry — as hypothesized in this paper and as we show more explicitly in a forthcoming paper.

10. In equation 40, I assume that sigma is not explicitly defined earlier. While its meaning (related to jet width) can be inferred, a brief definition would be helpful for clarity.

    You are right! We will add a piece of text to clarify the role of $\sigma_J$.

**Reviewer 2**

Firstly, I was not immediately struck by the originality, surprise, or novelty of the results. It seems to be a bit of an exercise in relatively straightforward applied mathematics with some linear model well-explored. It is then not so clear to me what I really did learn from the manuscript that I did not already know or could guess. It clearly does add colour and detail to our understanding of the forced, linearised barotropic vorticity equation applied in a beta- channel setting, but I am rather less convinced it actually helps us understand the atmosphere any better.

The authors point out some of the caveats of the setup, but I might add some further caveats: the BVE is linearised, but partial wave reflection at a jet edge in any dynamical setting would be accompanied by nonlinear filamentation. The authors point out that the jet is considered zonally symmetric but the real jet is importantly deformed by the nonlinear planetary waves that the paper is about. The detailed nature of the boundary conditions is important, but in reality this would vary on the zonal scale of the planetary waves themselves. The atmosphere is not barotropic, and would, at the least, involve a finite Rossby radius; after all, the compression of the column by the orography is the key forcing mechanism. Any nonlinearity would immediately introduce wave-wave interactions in which case the whole spectrum of the forcing becomes important. I am sure that further analyses could address some, perhaps all of these caveats, but that is not done in the present manuscript. The detailed results in the manuscript seem at face value strongly dependant on the rather strongly simplifying assumptions in the setup, but these same assumptions move the model quite far away from the observed atmosphere.

In summary, the details resulting from the analysis of a model with so many idealisations probably have rather limited applicability to the real atmosphere: which of your detailed conclusions would stand up to more realism?

If the idealised model actually solves a question we had about planetary waves then we learned something about the atmosphere using the model, but I do not believe that is the case in the present manuscript; at least that is not what is presented. As it stands the model analysis appears to just answer detailed questions about the model.

Would this type of analysis perhaps be better suited for an applied mathematics

`journal?`

The issues raised by the reviewer are all connected, so we try to answer them in one single swoop. The main criticism refers to the novelty of the results and the usefulness of the idealized model to understand the real world. Apparently, the reviewer is very knowledgeable and was able to anticipate most of our results. In fact, on a more general level, many papers in our science simply "add color and detail to our understanding", and whether this is worth an additional publication is a matter of judgment and context, which may be subjective and depend on the reader's background. Nevertheless, the reviewer's comment was very valuable to us, because it showed us that we failed to indicate the considerable value (as we think) of this work despite these potential issues.

During the past 12 years, there has been a surge of interest in Rossby wave resonance in the meteorological community following the work of Petoukhov et al. (2013) and Kornhuber et al. (2017). We think that part of that work is somewhat obscure and that some of the results are on rather thin ice. In particular, a key part of their approach is based on assumptions and approximations which are not satisfied in the real atmosphere (Wirth, 2020). At issue here is, in particular, the treatment of the meridional dimension. By contrast, our paper pays particular attention to the meridional dimension and the occurrence of meridional modes; at the same time, our method dispenses with some of the more questionable assumptions of the Petoukhov-Kornhuber approach. We therefore think that it is timely to revisit the issue of Rossby wave resonance using our less approximate method and, thereby, enhance our understanding for the underlying physics. Such a paper should be a valuable contribution at least for that part of the community which aims to diagnose observations in an attempt to quantify the role of resonant amplification in real episodes. And, importantly, this is why we think that our paper should not be published in an applied mathematics journal, but rather in a journal like WCD that is read by meteorologists.

Let us explain in a bit more detail. Most notably the work of Petoukhov et al. (2013, 2016) and Kornhuber et al. (2017) tries to diagnose Rossby wave resonance with a method that is based on the linearized barotropic vorticity equation plus a few additional approximations and ad-hoc rules. A crucial part of their method determines whether or not there is a zonal waveguide, and this boils down to considering the so-called refractive index and searching for two so-called turning latitudes. The underlying framework was pioneered in the work of Hoskins and Karoly (1981). Amongst others, in this framework one may well obtain more than one "resonant wavenumber", and this feature of the method was even used in order to define the "strength of a waveguide" (e.g., Fig. 1 in White and Admasu, 2025). As we argue in the current manuscript, these multiple resonant wavenumbers must be related to higher meridional modes. Therefore, it seems important to us to clarify the role of higher meridional modes, and this is indeed a major focus of our paper. The simple fact that we never obtained more than one resonant wavenumber with our (less approximate) method indicates a serious problem with the (more approximate) Petoukhov-Kornhuber approach. We think that this by itself is a novel result which is worth being discussed in a new publication.

In response to the reviewer's remarks we are planning to add a substantial amount of text to the introduction in order to talk about the Petoukhov-Kornhuber methodology and, thus, better motivate the current paper.

In addition, the current paper will gain added value in a few months time: we are planning to submit a follow-up study in which we explicitly compare results from the Petoukhov-

Kornhuber diagnostic approach with results from our own methodology. In that future paper, we will explicitly refer to the current manuscript, as the current results are valuable for understanding the results of that future paper. Obviously, we cannot assume that the reader of the current manuscript knows our plans for future publications; for this reason we will enhance our text in the conclusion section and thereby offer a sneak preview to our future plans.

Regarding the use of linear barotropic theory for investigating Rossby wave resonance, we plan to add a substantial amount of text to the introduction and the concluding section. We agree with almost everything the reviewer says, and yet we think that our paper makes a valuable contribution despite all these caveats. After all, linear resonance is a theoretical concept, and any theory implies idealization by making assumptions or approximations. This is the curse of theory, but this does not make theory obsolete. One possible way to proceed and to find out to what extent linear theory has value for understanding the real atmosphere in the present context would lie in repeating the analyses performed by Petoukhov, Kornhuber and the other quoted authors — except by using our method rather than theirs. We think that this would be a worthwhile exercise, but clearly this exercise transcends the scope of the present paper.

**References**

Fragkoulidis, G. and V. Wirth, 2020: Local Rossby wave packet amplitude, phase speed, and group velocity: Seasonal variability and their role in temperature extremes. *J. Climate*, **33(20)**, 8767–8787, DOI:10.1175/JCLI–D–19–0377.1.

Hoskins, B. J. and D. J. Karoly, 1981: The Steady Linear Response of a Spherical Atmosphere to Thermal and Orographic Forcing. *J. Atmos. Sci.*, **38**, 1179–1196.

Kornhuber, K., V. Petoukhov, S. Petri, S. Rahmstorf, and D. Coumou, 2017: Evidence for wave resonance as a key mechanism for generating high-amplitude quasi-stationary waves in boreal summer. *Climate Dynamics*, **49**, 1961–1979, doi:10.1007/s00,382–016–3399–6.

Petoukhov, V., S. Petri, S. Rahmstorf, D. Coumou, K. Kornhuber, and H. J. Schellnhuber, 2016: Role of quasiresonant planetary wave dynamics in recent boreal spring-to-autumn extreme events. *Proceedings of the National Academy of Sciences*, **113**, 6862–6867.

Petoukhov, V., S. Rahmstorf, S. Petri, and H.-J. Schellnhuber, 2013: Quasiresonant amplification of planetary waves and recent Northern Hemisphere weather extremes. *Proceedings of the National Academy of Sciences*, **110(14)**, 5336–5341, doi:10.1073/pnas.1222000,110.

White, R. H. and L. M. Admasu, 2025: Temporally and zonally varying atmospheric waveguides — climatologies and connections to quasi-stationary waves. *Weather and Climate Dynamics*, **6**, 549–570, https://doi.org/10.5194/wcd–6–549–2025.

Wirth, V., 2020: Waveguidability of idealized midlatitude jets and the limitations of ray tracing theory. *Weather Clim. Dynam.*, **1**, 111–125, https://doi.org/10.5194/wcd–1–111–2020.

---

## Author Response (AR1)

**Response to the editor's and the reviewers' comments**

In this document, the comments of the editor and the two reviewers are repeated in typewriter font, followed by our reply in normal font. We also mention in this document how we changed the manuscript in our attempt to account for the various comments. In addition, we provide a manuscript with track changes from the previous version to the revised version.

**Editor's remarks**

```
Dear Authors,

We have now received two reviews of your manuscript.  In light of these and based
on my own assessment, I would like to invite you to prepare your responses to
both reviews.  Please address in particular the comments made by reviewer no.
2 regarding the degree of novelty.

From a scientific perspective, I would like you to consider the following for
your discussion.  As you increase the sharpness of the jet, there is a possibility
of it becoming barotropically unstable.  In such cases, seeking stationary solutions
may not be meaningful.

Best wishes,

Sebastian Schemm
```

We thank the editor, Sebastian Schemm, for handling our paper and we are grateful for his remarks.

- Regarding the novelty of this work, we explicitly address this issue in our reply to reviewer 2.

- Regarding the possibility of barotropic instability: Yes, you are right, our strong jets are barotropically unstable according to Rayleight's inflection point theorem. We want to address your remark by making four points.

  - Even though the basic state flow may support barotropic instability, the stationary waves that we discuss in our paper are stable. The mathematical reason is that we only consider basic states that are westerly everywhere, and Howard's semicircle theorem then says that the phase speed of the barotropically unstable modes must be positive. Our focus on stationary modes then precludes instability. This is commented in the third paragraph of section 4.2.

  - In a more realistic framework aiming to represent the Earth's troposphere, one is more likely to obtain baroclinic instability rather than barotropic instability. Obviously, this is one of the fundamental limitations of the barotropic model: it is unable to represent baroclinic instability. Real jets in the real atmosphere occur in the presence of large gradients of the background Ertel potential vorticity (PV); however, this gradient is, to a large extent, due to a quasi-horizontal gradient in static stability rather than in relative vorticity. Again, since the barotropic model "does not know" anything about static stability, it cannot possibly represent this

feature of the real atmosphere. Despite this caveat, the linear barotropic model has frequently been used to study Rossby wave resonance, and in the current paper we follow this line of investigation; our focus is on an improved understanding of this model framework, especially as far as the meridional dimension is concerned.

– Previous investigations about Rossby wave resonance in observed episodes have usually focused on stationary waves, since stationary waves are more likely to be associated with extreme weather than traveling waves (Fragkoulidis and Wirth, 2020). This focus was achieved through time averaging, like, e.g., by analysing monthly means (Petoukhov et al., 2013) or by preprocessing with a 15-day running mean (Kornhuber et al., 2017). Therefore, we think that it makes sense to focus in our paper on stationary waves and their possible resonant amplification, and ignore the fact that the used model is unable to represent other phenomena (like baroclinic instability).

– Rossby wave resonance is typically discussed in the context of stationary waves, i.e., the resulting large-amplitude waves are produced by stationary forcing. By contrast, baroclinic instability does not require any forcing and the wave growth occurs through purely internal dynamics. These two issue are conceptually well separated, and formally they can be distinguished by their zonal phase speed (close to zero in the case of quasi-stationary waves, versus eastward in the case of baroclinic waves). In a linear framework, different wave modes with different phase speeds can be dealt with independently of each other. The focus on quasi-stationary waves (see previous item) is, therefore, mathematically well-posed.

We further addressed the issue by adding a footnote in section 4.2

**Reviewers' comments**

We thank the reviewers for carefully reading our manuscript and making insightful comments. Both reviewers mentioned that they enjoyed reading/reviewing the paper, and we are grateful for this particular feedback.

**Reviewer 1**

**Reply to his minor comments**

1. Typo corrected

2. In the Introduction, I would really appreciate the authors including a more detailed comparison with several relevant papers on resonance, such as Charney and DeVore, Tung and QRA, to better position this work in the existing literature.

   We agree that it is a good idea to provide more context regarding the previous literature about resonance. Accordingly, we added three new paragraphs (2nd paragraph, part of the 5th paragraph and the 6th paragraph in the introduction) in order to address this issue. Incidentally, these added pieces of text will also help to address the second reviewer's remark regarding the novelty of our work.

3. From equations 5 to 8, it seems that h hat is not dimensionless if D=500
   km, in equation 7, which appears inconsistent with the units in equation
   8.   I would suggest adjusting the delta function to have the dimension of
   $(length)^{-1}$.   Additionally, a brief explanation of the forcing expression of
   pseudo topography could be helpful for readers less familiar with this formulation.

   The parameter $D$ in (7) has dimensions of m, and the delta-function has dimensions of
   $m^{-1}$. Taken together, this renders $\hat{h}$ and, hence, $h'$ dimensionless. We added a sentence
   behind equation (7) to make this point explicit.

   Also, we modified our terminology and now talk about "pseudo-orography" instead of
   simply "orography", since the latter would be somewhat strange in a barotropic model.
   A sentence plus a reference was added right behind equation (5) to motivate the new
   terminology.

4. In Figure 1, the annotation for \meridional wind" need to be corrected to
   \zonal wind."

   Right, of course! Thank you for spotting this error. We corrected the figure caption
   accordingly.

5. I suggest adding more explanation regarding the psi value in equation 29,
   for example, by showing how it is derived by substituting back into equation
   17.

   That's a good idea! We enhanced our text before equation (29) and behind equation
   (30) in order account for the reviewer's comment.

6. When discussing the physical interpretation or the limitations of the n=0
   solution, either in the Results or Discussion sections, it may be helpful
   to explicitly note that the presence of singularities indicates that, beyond
   analyzing wavenumber, amplitude, and phase, the linear equations may no longer
   be valid for the solution.

   We are not sure whether we fully understand the reviewer's comment. To be sure, in
   the presence of resonance, wave amplitudes will grow linearly with time, and at some
   point linearization breaks down as a useful approximation. In the real world, nonlinear
   effects (amongst others) start to play a role, and this may lead to the vanishing of the
   singular modes. In any case, we are aware that the results from linear theory must be
   taken with a grain of salt. However, this does not reduce the value of linear analysis to
   zero, and linear theory has been applied to diagnose Rossby wave resonance often times
   in the past. In fact, a thorough understanding of linear resonance may prove useful to
   understand nonlinear effects including multiple equilibria.

   As a response, we added an new paragraph to the concluding section (beginning with
   "One may question the hole idea of using a linear barotropic model....")  in order to
   explicitly address this issue. This new paragraph points to the obvious limitations of
   linear theory and clarifies our primary aim, which is to better understand recent papers
   that do use the linear barotropic model.

7. Typo corrected.

8. Typo corrected.

9. However, it seems that a zonal resonance solution (n=0) is not necessarily
   impossible in realistic conditions.  For example, in line 557, the jet corresponding

to the n=0 solution is described as \unrealistically wide," yet Figure 6
still shows a solution with Ly=1500km, though its amplitude is weaker.  Some
clarification of this point could strengthen the discussion.

You are right: for a constant basic state wind with partly or fully transparent boundaries
one obtains the $n = 0$ resonance even for a rather narrow domain (as exemplified in our
Fig. 6). However, this sharp peak is only generated from that part of the solution, which
is able to penetrate the boundaries; this part of the solution is a plane wave with purely
zonal propagation, which implies that the effective width of the channel is infinite (for
this part of the solution). Therefore, its mechanism is fundamentally different from
the mechanism for the higher meridional modes $n = 1, 2, \ldots$, which require meridional
propagation and (partial) reflection in the meridional direction.

We enhanced the section containing this issue (the bullet in the summary section start-
ing with "Even for the extreme case ...") in order to better indicate what we mean and
to make a connection with what we said earlier in section 3.3. ("We emphasize, again,
the different nature of the two resonant peaks ..."). At the same time, we chose to keep
the discussion short, because this peak does not have an analog in spherical geometry
— as hypothesized in this paper and as we show more explicitly in a forthcoming paper.

10. In equation 40, I assume that sigma is not explicitly defined earlier.  While
    its meaning (related to jet width) can be inferred, a brief definition would
    be helpful for clarity.

    You are right! We added half a sentence to clarify the role of $\sigma_J$.

**Reviewer 2**

Firstly, I was not immediately struck by the originality, surprise, or novelty
of the results.  It seems to be a bit of an exercise in relatively straightforward
applied mathematics with some linear model well-explored.  It is then not so clear
to me what I really did learn from the manuscript that I did not already know
or could guess.  It clearly does add colour and detail to our understanding of
the forced, linearised barotropic vorticity equation applied in a beta- channel
setting, but I am rather less convinced it actually helps us understand the atmosphere
any better.

The authors point out some of the caveats of the setup, but I might add some further
caveats:  the BVE is linearised, but partial wave reflection at a jet edge in
any dynamical setting would be accompanied by nonlinear filamentation.  The authors
point out that the jet is considered zonally symmetric but the real jet is importantly
deformed by the nonlinear planetary waves that the paper is about.  The detailed
nature of the boundary conditions is important, but in reality this would vary
on the zonal scale of the planetary waves themselves.  The atmosphere is not barotropic,
and would, at the least, involve a finite Rossby radius; after all, the compression
of the column by the orography is the key forcing mechanism.  Any nonlinearity
would immediately introduce wave-wave interactions in which case the whole spectrum
of the forcing becomes important.  I am sure that further analyses could address
some, perhaps all of these caveats, but that is not done in the present manuscript.
The detailed results in the manuscript seem at face value strongly dependant on

the rather strongly simplifying assumptions in the setup, but these same assumptions
move the model quite far away from the observed atmosphere.

In summary, the details resulting from the analysis of a model with so many idealisations
probably have rather limited applicability to the real atmosphere:  which of your
detailed conclusions would stand up to more realism?

If the idealised model actually solves a question we had about planetary waves
then we learned something about the atmosphere using the model, but I do not believe
that is the case in the present manuscript; at least that is not what is presented.
As it stands the model analysis appears to just answer detailed questions about
the model.

Would this type of analysis perhaps be better suited for an applied mathematics
journal?

The issues raised by the reviewer are all connected, so we try to answer them in one single
swoop. The main criticism refers to the novelty of the results and the usefulness of the
idealized model to understand the real world.

During the past 13 years, there has been a surge of interest in Rossby wave resonance in the
meteorological community following the work of Petoukhov et al. (2013) and Kornhuber et al.
(2017). We think that part of that work is somewhat obscure and some of the results are on
thin ice. In particular, the authors make a number of assumptions and approximations which
seem not to be satisfied in the real atmosphere (Wirth, 2020). At issue here is, in particular,
the treatment of the meridional dimension. A main aspect of our results which was novel
to us and thus we think is novel to the community is the limitation of resonance to the
first meridional mode owing to the narrowness of the jet. In addition, our method dispenses
with some of the more questionable assumptions of the Petoukhov-Kornhuber approach.
Therefore, we believe that it is timely to revisit the issue of Rossby wave resonance with a
less approximate method and enhance our understanding for the underlying physics. Such a
paper should be valuable for at least for that part of the community which aims to diagnose
observations in an attempt to quantify the role of resonant amplification in real episodes.
And, importantly, this is why we think that our paper should not be published in an applied
mathematics journal, but rather in a journal like WCD that is read by meteorologists.

Let us explain in a bit more detail. Most notably the work of Petoukhov et al. (2013, 2016)
and Kornhuber et al. (2017) tries to diagnose Rossby wave resonance with a method that is
based on the linearized barotropic vorticity equation plus a few additional approximations
and ad-hoc rules. A crucial part of their method determines whether or not there is a zonal
waveguide, and this boils down to considering the so-called refractive index and searching
for two so-called turning latitudes. The underlying framework was pioneered by Hoskins
and Karoly (1981). Amongst others, in this framework one may well obtain more than one
"resonant wavenumber", and this feature of the method was even used in order to define
the "strength of a waveguide" (e.g., Fig. 1 in  White and Admasu, 2025). As we argue
in the current manuscript, these multiple resonant wavenumbers must be related to higher
meridional modes. Therefore, it seems important to us to clarify the role of the meridional
modes, and this is indeed a major focus of our paper.

In response to the reviewer's remarks we added a substantial amount of text to the introduc-
tion in order to talk about the Petoukhov-Kornhuber methodology and, thus, better motivate

the current paper. In addition, we supplemented the title of the paper by the phrase "...: towards a better understanding of the meridional wave structure". We do think that our paper, and specifically our focus on the meridional structure of the solution, adds interesting and novel information to the discussion. At the same time, the reviewer's comment was very valuable to us, because it showed us that we failed to get this message across in the original version.

Furthermore, the current paper will gain added value in a few months time: we are planning to submit a follow-up study in which we explicitly compare results from the Petoukhov-Kornhuber diagnostic with results from our own methodology. In that future paper, we will explicitly refer to the current manuscript, as the current results will help to understand the results of that future paper. Obviously, the reader of the current manuscript cannot possibly anticipate our plans for future publications; for this reason we enhanced our text in the conclusion section and now offer a sneak preview to our future plans (second-to-last paragraph in the conclusions section).

Regarding the use of linear barotropic theory for investigating Rossby wave resonance, we added a substantial amount of text to the concluding section ("One may question the whole idea...."). We agree with almost everything the reviewer says, and yet we think that our paper makes a valuable contribution despite all these caveats. After all, linear resonance is a theoretical concept, and any theory implies idealization by making assumptions or approximations. This is the curse of theory, but this does not make theory obsolete. One possible way to proceed and to find out to what extent the linear theory has value for understanding the real atmosphere would lie in repeating the analyses performed by Petoukhov, Kornhuber and the other quoted authors — except by using our method rather than theirs. We think that this is a worthwhile exercise, but clearly this exercise transcends the scope of the present paper.

**References**

Fragkoulidis, G. and V. Wirth, 2020: Local Rossby wave packet amplitude, phase speed, and group velocity: Seasonal variability and their role in temperature extremes. *J. Climate*, **33(20)**, 8767–8787, DOI:10.1175/JCLI–D–19–0377.1.

Hoskins, B. J. and D. J. Karoly, 1981: The Steady Linear Response of a Spherical Atmosphere to Thermal and Orographic Forcing. *J. Atmos. Sci.*, **38**, 1179–1196.

Kornhuber, K., V. Petoukhov, S. Petri, S. Rahmstorf, and D. Coumou, 2017: Evidence for wave resonance as a key mechanism for generating high-amplitude quasi-stationary waves in boreal summer. *Climate Dynamics*, **49**, 1961–1979, doi:10.1007/s00,382–016–3399–6.

Petoukhov, V., S. Petri, S. Rahmstorf, D. Coumou, K. Kornhuber, and H. J. Schellnhuber, 2016: Role of quasiresonant planetary wave dynamics in recent boreal spring-to-autumn extreme events. *Proceedings of the National Academy of Sciences*, **113**, 6862–6867.

Petoukhov, V., S. Rahmstorf, S. Petri, and H.-J. Schellnhuber, 2013: Quasiresonant amplification of planetary waves and recent Northern Hemisphere weather extremes. *Proceedings of the National Academy of Sciences*, **110(14)**, 5336–5341, doi:10.1073/pnas.1222000,110.

White, R. H. and L. M. Admasu, 2025: Temporally and zonally varying atmospheric waveguides — climatologies and connections to quasi-stationary waves. *Weather and Climate Dynamics*, **6**, 549–570, https://doi.org/10.5194/wcd–6–549–2025.

Wirth, V., 2020: Waveguidability of idealized midlatitude jets and the limitations of ray tracing theory. *Weather Clim. Dynam.*, **1**, 111–125, https://doi.org/10.5194/wcd–1–111–2020.